# Analysis of Arctic Spring Ozone Anomaly in the Phases of QBO and 11-Year Solar Cycle for 1979–2017

**Yousuke Yamashita** [1,2], **Hideharu Akiyoshi** [1,*] **and Masaaki Takahashi** [1,3]

1   Earth System Division, National Institute for Environmental Studies, Tsukuba 305-8506, Japan;
    yamashita.yosuke@nies.go.jp (Y.Y.); masaaki@aori.u-tokyo.ac.jp (M.T.)
2   Japan Agency for Marine-Earth Science and Technology, Yokohama 236-0001, Japan
3   Atmosphere and Ocean Research Institute, The University of Tokyo, Kashiwa 277-8564, Japan
*   Correspondence: hakiyosi@nies.go.jp; Tel.: +81-29-851-4732

**Abstract:** Arctic ozone amount in winter to spring shows large year-to-year variation. This study investigates Arctic spring ozone in relation to the phase of quasi-biennial oscillation (QBO)/the 11-year solar cycle, using satellite observations, reanalysis data, and outputs of a chemistry climate model (CCM) during the period of 1979–2017. For this duration, we found that the composite mean of the Northern Hemisphere high-latitude total ozone in the QBO-westerly (QBO-W)/solar minimum ($S_{min}$) phase is slightly smaller than those averaged for the QBO-W/$S_{max}$ and QBO-E/$S_{max}$ years in March. An analysis of a passive ozone tracer in the CCM simulation indicates that this negative anomaly is primarily caused by transport. The negative anomaly is consistent with a weakening of the residual mean downward motion in the polar lower stratosphere. The contribution of chemical processes estimated using the column amount difference between ozone and the passive ozone tracer is between 10–20% of the total anomaly in March. The lower ozone levels in the Arctic spring during the QBO-W/$S_{min}$ years are associated with a stronger Arctic polar vortex from late winter to early spring, which is linked to the reduced occurrence of sudden stratospheric warming in the winter during the QBO-W/$S_{min}$ years.

**Keywords:** Arctic ozone; QBO; 11-year solar cycle; sudden stratospheric warming; chemistry-climate model

## 1. Introduction

During winter and spring, the Arctic ozone exhibits a significant interannual variation [1]. This variation is associated with the variation in the Northern Hemisphere (NH) polar vortex. When the polar vortex is strong and stable, the temperature inside the vortex is lower than climatology. This results in greater chemical ozone destruction due to increased formation of polar stratospheric clouds. Additionally, the ozone transport from midlatitudes to the vortex is reduced.

The quasi-biennial oscillation (QBO) in the equatorial stratosphere and the 11-year solar cycle are known to cause interannual variability in the NH polar vortex (e.g., [2–9]). Holton and Tan [2] found that the NH polar vortex is anomalously strong during the westerly phase of the QBO (QBO-W), defined at 50 hPa for early and late winter, whereas it is anomalously weak in the easterly phase (QBO-E). In addition to the QBO, the NH polar vortex may vary with the influence of the solar maximum ($S_{max}$) and solar minimum ($S_{min}$) of the 11-year solar cycle. Labitzke and van Loon [5] and others (e.g., Naito and Hirota [6]) demonstrated that the NH polar vortex is strong in early winter and weak with frequent sudden stratospheric warming episodes in late winter during QBO-W/$S_{max}$ conditions. Yamashita et al. [10] supported the Labitzke and van Loon [5] study by the composite analysis of the QBO and solar cycle with four groups (QBO-W/$S_{max}$, QBO-W/$S_{min}$, QBO-E/$S_{max}$, and QBO-E/$S_{min}$), and indicated that the strengthened polar vortex in early winter is followed by a weakened polar vortex in late winter for the QBO-W/$S_{max}$

group. It is noteworthy that Kren et al. [11] suggests by their detailed analysis of the Whole Atmosphere Community Climate Model (WACCM) simulations that the apparent relationship between the QBO, the solar cycle, and the state of the NH polar vortex can arise from sampling over relatively short periods.

The relationship of the NH polar vortex strength with the phases of the QBO and the 11-year solar cycle may change from early winter to late winter (or early spring). As ozone depletion is enhanced in a stable and cold polar vortex from late winter to early spring, it is necessary to know the QBO phase wherein the NH polar vortex is strongest during late winter and early spring when sunlight reaches the Arctic region. For the QBO-W/$S_{min}$ group, Labitzke and van Loon [5] indicated that the NH polar vortex is strong with less sudden stratospheric warming in late winter. Camp and Tung [12] suggested that the NH polar vortex is the most stable and least perturbed in late winter under the QBO-W/$S_{min}$ group relative to the other three groups (QBO-W/$S_{max}$, QBO-E/$S_{max}$, and QBO-E/$S_{min}$), in agreement with the work of Labitzke and van Loon [5]. These results imply that minimum ozone levels may occur under the QBO-W/$S_{min}$ group. Furthermore, Li and Tung [13] found that the Arctic total ozone in March is the lowest in magnitude for the QBO-W/$S_{min}$ group by the analysis of the TOMS/OMI observations. Therefore, a quantitative estimation of the influence of the ozone transport and the chemical ozone destruction on the amount of the Arctic ozone for the QBO-W/$S_{min}$ years remains a challenging issue for explaining the low total ozone levels in the late winter and early spring.

In this study, we analyze the Arctic ozone using TOMS/OMI observations, reanalysis data, and the outputs of a chemistry climate model (CCM). The anomaly from the average Arctic total ozone for all years during 1979–2011 is calculated for the QBO-W/$S_{min}$ group during late winter and early spring. We estimate the amount of ozone transport and chemical ozone destruction, and their effects on the derived Arctic ozone anomaly for the QBO-W/$S_{min}$ group using the model output. Additionally, we analyze the occurrence of sudden stratospheric warming during the QBO/solar phases and investigate relationships between sudden stratospheric warming, QBO/solar phases, polar vortex strength, and ozone amount during the seasonal evolution from winter to spring.

## 2. Materials and Methods

### 2.1. Data and Model Description

The daily mean total ozone data from version 8 of the NIMBUS-7/Total Ozone Mapping Spectrometer (TOMS) (1979–1993, $1.25° \times 1.0°$ longitude–latitude grids), EP/TOMS (1996–2004, $1.25° \times 1.0°$ longitude–latitude grids), and Ozone Monitoring Instrument (OMI) (2005–2017, $0.25° \times 0.25°$ longitude–latitude grids) satellites are analyzed. These satellite observations are henceforth referred to as TOMS/OMI satellite data. We also analyze the daily total ozone data from the European Centre for Medium-Range Weather Forecasts (ECMWF) Interim Re-Analysis (ERA-Interim) for 1979–2017 [14]. The data set with $2.5° \times 2.5°$ longitude–latitude grids provided by the ECMWF is used. In order to use the same grid points as those of the ERA-Interim, the TOMS/OMI satellite data are averaged and interpolated to $2.5° \times 2.5°$ longitude–latitude grids.

The potential vorticity (PV) data at 430 K and 475 K levels are also obtained from the ERA-Interim reanalysis dataset with a resolution of $2.5° \times 2.5°$ in longitude and latitude. The PV values at 430 K and 475 K levels are interpolated to the PV values at the 450 K level, and subsequently used to calculate equivalent latitudes [15]. Thereafter, the equivalent latitudes are used to analyze the total ozone from the TOMS/OMI satellite data.

The daily mean ozone data from the Center for Climate System Research/National Institute for Environmental Studies (CCSR/NIES)-MIROC3.2 CCM are also analyzed. This model has a T42 horizontal resolution with 34 vertical levels in the sigma–p hybrid coordinate system [16]. The top layer is located at approximately 80 km (0.01 hPa). The 11-year solar cycle effect and QBO are included in the CCM [17].

The model output data used in the analysis are those from the REF-C1SD experiment of the International Global Atmospheric Chemistry/Stratosphere-troposphere Processes

and their Role in the Climate CCM Initiative (CCMI) [18]. The settings for the REF-C1SD experiment are the same as those for the REF-C1 experiment: The experiment includes the historical evolution of the sea surface temperature (SST), sea ice, solar cycle, volcanic aerosol, greenhouse gas (GHG) concentrations, and ozone-depleting substance (ODS) concentrations. The HadISST-1 data set provided by the UK Met Office Hadley Centre [19] is used for the historical evolution of SST and sea ice. The daily mean data of the spectral solar irradiance from the NRLSSI model [20] are used for each radiation bin of the model. The evolution of GHG and ODS concentrations are from the RCP-historical scenario and the World Meteorological Organization (WMO) baseline (A1) scenario [21], respectively.

In contrast to the REF-C1 experiment, the REF-C1SD experiment uses the hindcast simulation, wherein the observational data are assimilated into the meteorological fields of the model. Here, nudging is used as an assimilation method: the zonal wind, meridional wind, and temperature fields of the model are nudged towards those of the ERA-Interim data set with a 6-h interval from the surface to the 1-hPa level. Above the 1 hPa level, the climatological zonal mean zonal wind and temperature fields of the CIRA86 [22] are used as nudging data due to the lack of ERA-Interim data. The REF-C1SD experiment is run from 1 January 1979 to 31 December 2011, using the output of the REF-C1 experiment as the initial data.

To diagnose the influence of ozone transport on the change in ozone, a passive ozone tracer that is advected without changing due to chemistry is included in the model. Calculating a passive ozone tracer in the model is not straightforward because the ozone concentration in our model is a diagnostic quantity evaluated from a prognostic variable of odd oxygen ($O_x = O + O_3$) on the assumption of photochemical equilibrium. In this study, we consider the passive odd oxygen tracer as a passive ozone tracer. This is because atomic oxygen concentration is very low in the middle and lower stratosphere, and thus, the ozone concentration is nearly equal to the odd oxygen concentration. We assume that the difference between chemically active odd oxygen and the passive odd oxygen tracer is nearly equal to that between chemically active ozone and the passive ozone tracer, and, thus, the chemical anomaly of ozone.

The initialization for the passive tracer was performed for 1 December and 1 June each year by setting the concentration of the passive odd oxygen tracer to that of the chemically active odd oxygen. With the time-interval of 6 months for initialization, the passive tracer concentration in the upper stratosphere is different extensively from the chemically controlled, observed odd-oxygen distribution by the time for the next initialization or shortly after the initialization. This is due to the short time scale of odd oxygen chemical reactions in the upper stratosphere and lower mesosphere above 10 hPa, especially in February and March. Subsequent evaluation of the transport effect in the tracer distribution that is unrealistically far from the observed distribution is problematic. As we focus on an altitude range where the ozone amount accounts for most of the total ozone, the odd oxygen tracer function is specified only around these altitudes in the model, where the time scale of an ozone chemical reaction is half a month to more than 1 year [23]. That is, the passive odd oxygen tracer is not chemically altered in the vertical range of 220 hPa to 12 hPa in the model, not related to diabatic heating calculations in all regions of the CCM and influenced by transport in the same manner as chemically active odd oxygen. Subsequently, we can observe the chemical and transport anomalies within that altitude range, which may largely explain the total ozone anomalies.

### 2.2. Grouping of Years According to the Phases of the QBO and Solar Cycle and Definition of the Equivalent Latitude

We define the phases of the QBO and the solar cycle, and subsequently group the years according to the phases using the same method as Yamashita et al. [10], where equatorial zonal mean zonal wind at 50 hPa is used for the grouping. Previous studies have used this pressure level, or those slightly above 40 hPa for the grouping [3–6]. Figure 1a shows the yearly time-series of the December–January–February (DJF) mean of the equatorial zonal mean zonal wind at 50 hPa and the result of the grouping into 4-phase groups for each year.

Figure 1b shows the DJF mean of 10.7 cm solar radio flux (F10.7) with the grouping. The F10.7 is smoothed with the Fourier filter of a 2.5-year rectangular window in advance. The data is standardized by subtracting the average and dividing by the standard deviation. Resultantly, five years of the 1979–2011 period are categorized as the QBO-W/$S_{min}$ group (1986, 2005, 2007, 2009, and 2011), as shown by the closed triangles in Figure 1. Years of the other groups are as follows; the QBO-W/$S_{max}$ group: 1981, 1983, 1989, and 2000; the QBO-E/$S_{max}$ group: 1980, 1982, 1990, 1999, and 2002; the QBO-E/$S_{min}$ group: 1985, 1997, 2004, 2006, and 2008.

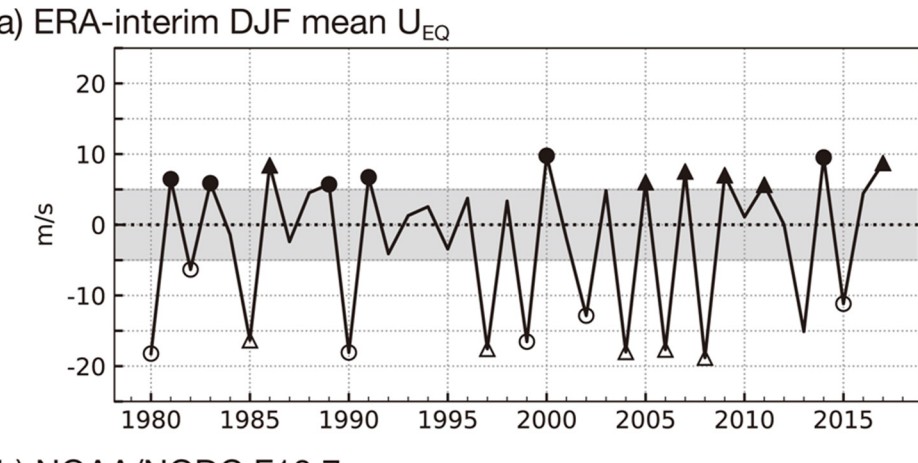

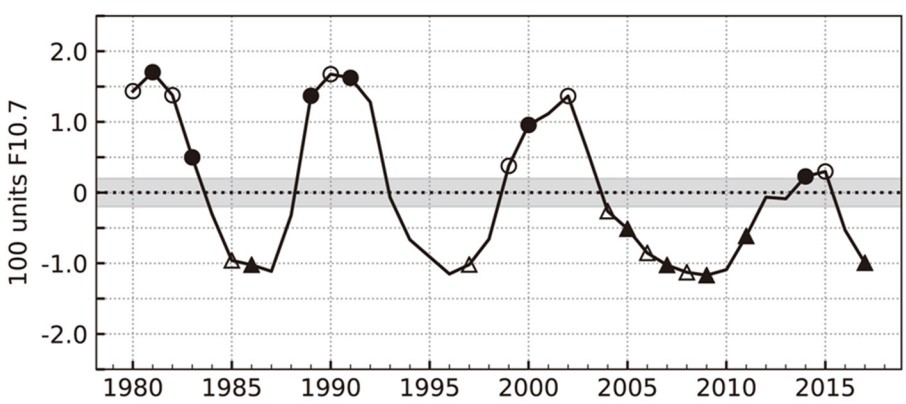

**Figure 1.** (**a**) Yearly time-series of the DJF mean zonal-mean zonal wind over 10° S–10° N at 50 hPa from ERA-Interim data. Absolute values of the equatorial zonal wind speed less than 5 m/s are gray-shaded; (**b**) Same as (**a**), but for the DJF mean of F10.7 from NOAA/NGDC. F10.7 is smoothed using a Fourier filter of a 2.5-year rectangular window, prior to calculating monthly mean values. Absolute values of F10.7 less than 0.2 are gray-shaded. Closed/open symbols denote QBO-W/QBO-E. Circles/triangles denote $S_{max}/S_{min}$. The closed triangles denote the QBO-W/$S_{min}$ years, open triangles denote the QBO-E/$S_{min}$ years, and open and closed circles denote the QBO-W/$S_{max}$ and QBO-E/$S_{max}$ years, respectively.

The years after 2011 are also included in this figure for reference, although REF-C1SD simulations were not performed for these years. The year 2014 is categorized into the QBO-W/$S_{max}$ group, 2015 into the QBO-E/$S_{max}$ group, and 2017 into the QBO-W/$S_{min}$ group. The years 2012, 2013, and 2016 are in the grey zones of the figures.

As opposed to the Southern Hemisphere, planetary wave activities in the NH have large magnitudes, and the longitudinal variation of the polar vortex area is also high. This causes a significant longitudinal variation in the polar vortex, inducing difficulties in analyzing the ozone change in the polar vortex area using a geographical latitude. Hence,

we use the equivalent latitude [15] instead of the geographical latitude. In this study, the equivalent latitude was defined by the PV values on a 450-K isentropic surface.

## 3. Results and Discussion

*3.1. A Scatter Plot for the Arctic Total Ozone against the QBO and Solar Cycle Phases in March*

Figure 2a shows the total ozone observed by TOMS averaged over the grids within the equivalent latitudes from 70° N to 90° N in March against the DJF mean of zonal mean zonal wind over 10° S–10° N at 50 hPa from ERA-Interim reanalysis data and the DJF mean of F10.7. The QBO-W/$S_{min}$ group (1986, 2005, 2007, 2009, and 2011) is located at the bottom right of the figure (white portion). The total ozone of the OMI in 2017, which is the only QBO-W/$S_{min}$ year after 2011, also shows a lower ozone value of 392.5 DU, as indicated in light blue color, compared to the average value of 410.1 DU for the years 1980–2017. Figure 2b is the same as Figure 2a, but for total ozone from ERA-Interim data. In this figure, the years 1994, 1995, and 1996 are included, which are missing in the TOMS data and are not categorized into any of the four groups. The total ozone amount is nearly the same between these two datasets. Note that the total ozone of ERA-Interim data for 2017 does not indicate a negative anomaly, whereas that of OMI data indicates a slight negative anomaly. Figure 2c is the same as Figure 2a but for the results of the REF-C1SD experiment of the CCM. Note that the years after 2011 are not included in our analysis in Figure 3, Figures 4–14, as well as in Figure 2c, since the REF-C1SD simulation ended in 2011. Although there is a minor difference in the absolute amount between the CCM results (Figure 2c) and observations (Figure 2b), the CCM satisfactorily simulates the anomalies of total ozone in March.

The total ozone in February is shown in Figure S1 in the Supplement Material. Generally, the total ozone in February is slightly lower than that in March. The figure indicates that the CCM satisfactorily simulates the total ozone in February as well.

*3.2. Arctic Total Ozone and Polar Night Jet in March in the QBO and 11-Year Solar Cycle Phases*

Figure 3 shows the monthly mean total ozone in the Arctic for March averaged for the years in the QBO and 11-year solar cycle phases and their standard deviations. Panels (a), (b), and (c) correspond to those in Figure 2, i.e., the results from TOMS/OMI, ERA-interim, and REF-C1SD. Because of the large standard deviations compared to the mean value differences, it is hard to find significant differences in the total ozone amount among the categories; the total ozone amount in the QBO-W/$S_{min}$ years indicates 42 DU (~10%) smaller than in the average of QBO-W/$S_{max}$ years and 46 DU (~11%) smaller than in the average of QBO-E/$S_{max}$ years for the TOMS/OMI data (panel (a)) and CCM (panel (c)), although it is not statistically significant at the 68% confidence level (1 sigma). However, most part of the upper half of the error bar of the QBO-W/$S_{max}$ years is located above the error bar of the QBO-W/$S_{min}$ years and the lower half of the error bar of the QBO-W/$S_{min}$ years is located below the error bar of the QBO-W/$S_{max}$ years. A similar situation is evident between the QBO-W/$S_{min}$ years and the QBO-E/$S_{max}$ years in panels (a) and (c). The total ozone in the QBO-E/$S_{min}$ years in panels (a), (b), and (c) and that of the ERA-Interim data in the QBO-E/$S_{max}$ years in panel (b) have very large standard deviations; hence, it is difficult to compare these total ozone amounts to that in the QBO-W/$S_{min}$ years. Furthermore, comparisons of the total ozone amount between QBO-W and QBO-E and between $S_{max}$ and $S_{min}$ are also difficult owing to the large standard deviation and small mean value differences.

A similar situation is evident in February that the average total ozone for the QBO-W/$S_{min}$ years are smaller than those for the other groups while the range of the standard deviation overlaps with those of the other groups (Figure S2).

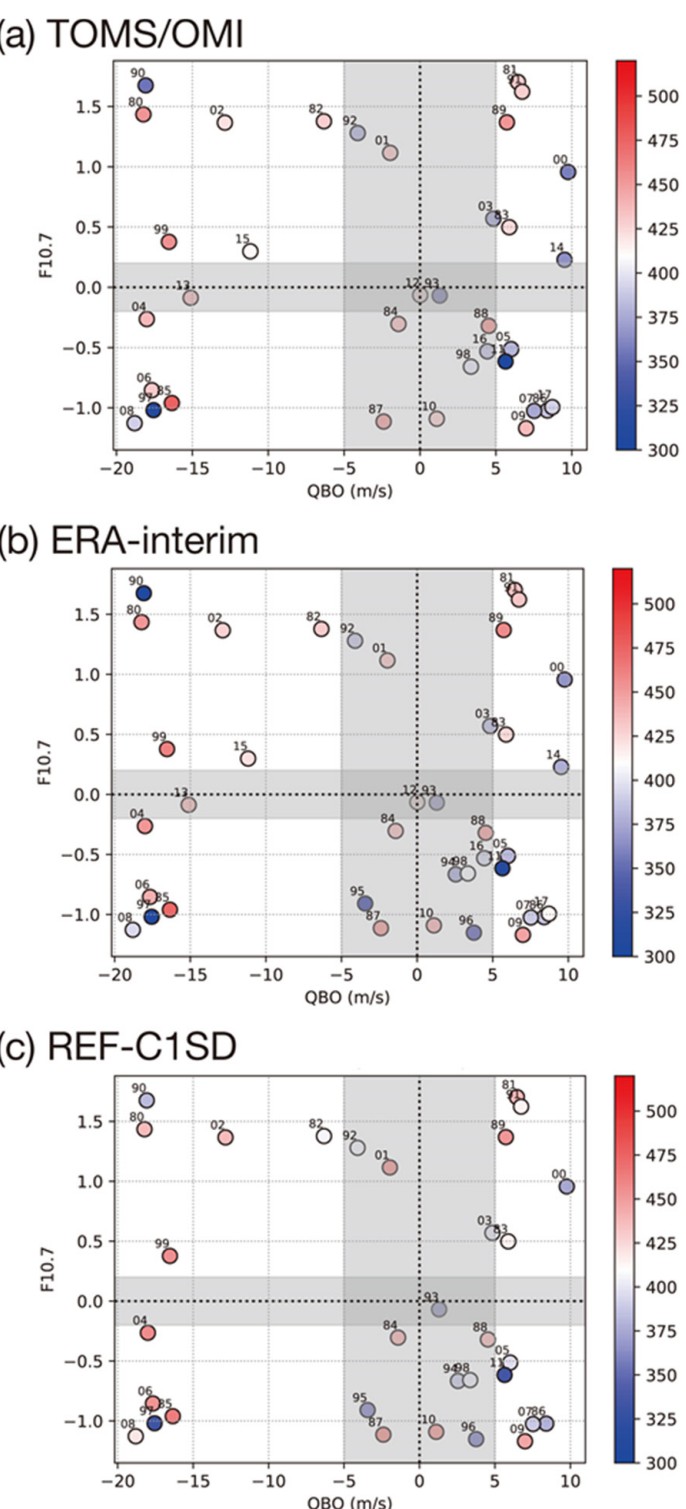

**Figure 2.** (**a**) Scatter plot between the QBO and F10.7 for the total ozone in March (color) averaged over the equivalent latitude from 70° N to 90 N for (**a**) TOMS/OMI observations, (**b**) ERA-Interim reanalysis data, and (**c**) REF-C1SD experiments. The average values of the TOMS/OMI observations, ERA-Interim reanalysis data, and REF-C1SD experiments are 410.1 DU, 408.4 DU, and 415.0 DU, respectively, which correspond to white on the color scale.

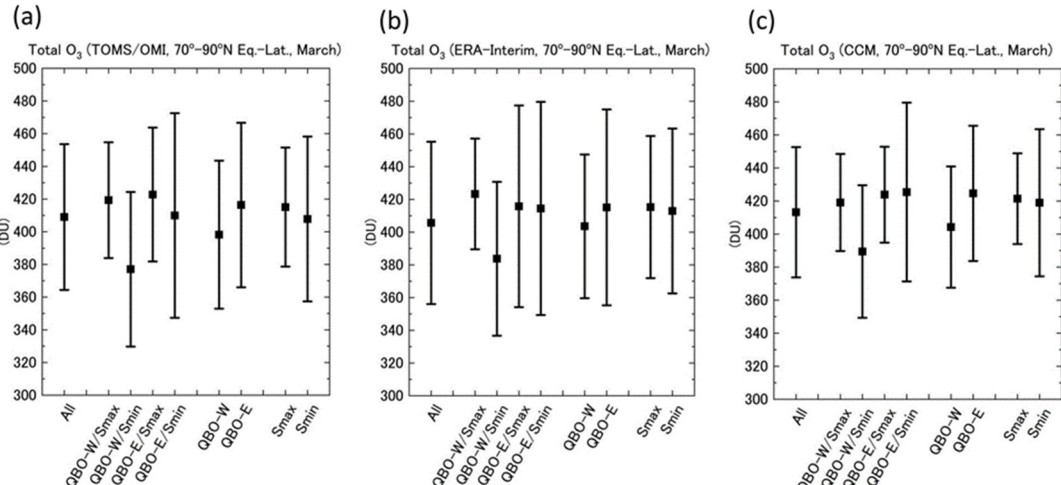

**Figure 3.** Monthly mean total ozone averaged over 70−90 N equivalent latitudes in March with units of dobson. The average values for all the years 1980–2011, the four QBO/solar activity phase groups, the two QBO phase groups, and the two solar activity phase groups are indicated by squares with the standard deviations. (**a**) TOMS/OMI data, (**b**) ERA-Interim data, and (**c**) CCM output.

Figure 4 shows the monthly mean zonal-mean zonal wind over 50–70° N at 10 hPa and 50 hPa in March averaged for the years in the QBO and 11-year solar cycle phases and their standard deviations, for which the ERA-Interim reanalysis data are used. It is expected that a stronger zonal-mean zonal wind, i.e., a stronger polar vortex, is associated with a lower total ozone because of less ozone transport to the polar regions across the polar vortex boundary and more chemical ozone destruction due to the lower temperature. The upper part of the error bar of the QBO-W/$S_{min}$ years is located above the error bar of the QBO-W/$S_{max}$ years; however, the lower part of the error bar of the QBO-W/$S_{max}$ years is not below the error bar of the QBO-W/$S_{min}$ years. The QBO-E/$S_{max}$ years and the QBO-E/$S_{min}$ years have large standard deviations, hence, comparisons with the QBO-W/$S_{min}$ years are difficult. The February mean zonal-mean zonal wind also did not indicate the larger magnitude for the QBO-W/$S_{min}$ years compared to the other groups, considering the range of the standard deviations (Figure S3).

These comparisons suggest that the total ozone amount in March averaged for the QBO-W/$S_{min}$ years is slightly smaller than those averaged for the QBO-W/$S_{max}$ and QBO-E/$S_{max}$ years, while it is difficult to say that the zonal-mean zonal wind averaged for the QBO-W/$S_{min}$ years is larger than those of the other phases. Although comparison of the total ozone between the QBO-W/$S_{min}$ years and the climatology (all the years) is not as evident as that between the QBO-W/$S_{min}$ years and the QBO-W/$S_{max}$ years or the QBO-E/$S_{max}$ years, hereafter, we show anomalies of ozone and the associated dynamical field of the QBO-W/$S_{min}$ years from the climatology.

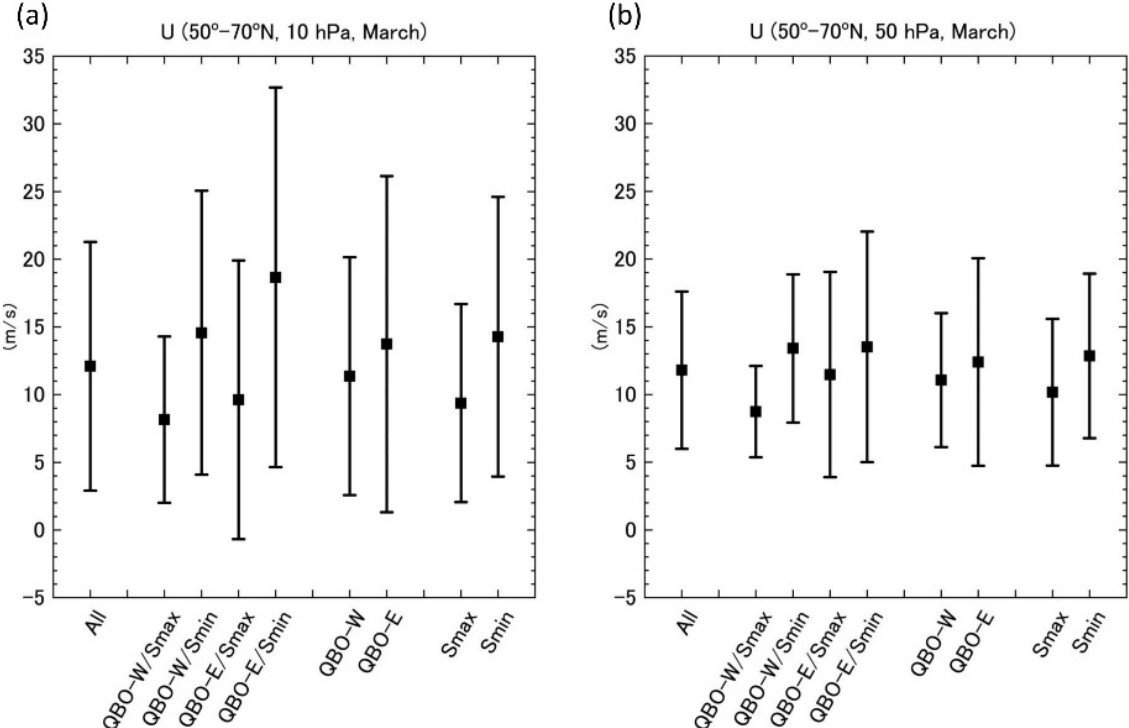

**Figure 4.** Monthly mean, zonal mean zonal wind averaged over 50−70 N at 10 hPa (**a**) and 50 hPa (**b**) in March from ERA-Interim data with units of m/s. The average values for all the years 1980–2011, the four QBO/solar activity phase groups, the two QBO phase groups, and the two solar activity phase groups are indicated by squares with the standard deviations.

### 3.3. Zonal Mean Zonal Wind Evolution in the Stratosphere of the QBO-W/$S_{min}$ Years and Associated Dynamical Fields

As explained in Section 2.2, five years (1986, 2005, 2007, 2009, and 2011) between 1979–2011 are categorized as belonging to the QBO-W/$S_{min}$ group. Note that, among these years, the total ozone in March 2009 is exceptionally large in magnitude (Figure 2). Herein, the daily time-series of the zonal mean zonal wind over 50–70° N at 10 hPa and 50 hPa, related to the polar vortex intensity, are shown for these five years of the QBO-W/$S_{min}$ group (solid lines in Figure 5). The ERA-Interim reanalysis data are used for the figure. The broken line denotes the daily zonal wind from the average of all years (1979–2011), smoothed by a Fourier filter with a rectangular window of three months. There were slightly strong polar vortices (that is, stronger westerly winds) during December–February for four of the QBO-W/$S_{min}$ years excluding 2009, compared to the average of 1979–2011. The years 1986, 2005, and 2011 dominantly contributes to the stronger westerly wind of the QBO-W/$S_{min}$ years from mid-February to early March in Figure 5. The lower panel of Figure 5 is the same as that of the upper panel but for 50 hPa, where large chemical ozone loss occurs in winter and the ozone anomaly around this pressure level greatly contributes to total ozone anomaly. The time evolution of each year is similar to that at 10hPa, but slightly delayed. The zonal mean zonal wind for the four years excluding 2009 tends to indicate stronger westerlies than the climatology, as in 10 hPa.

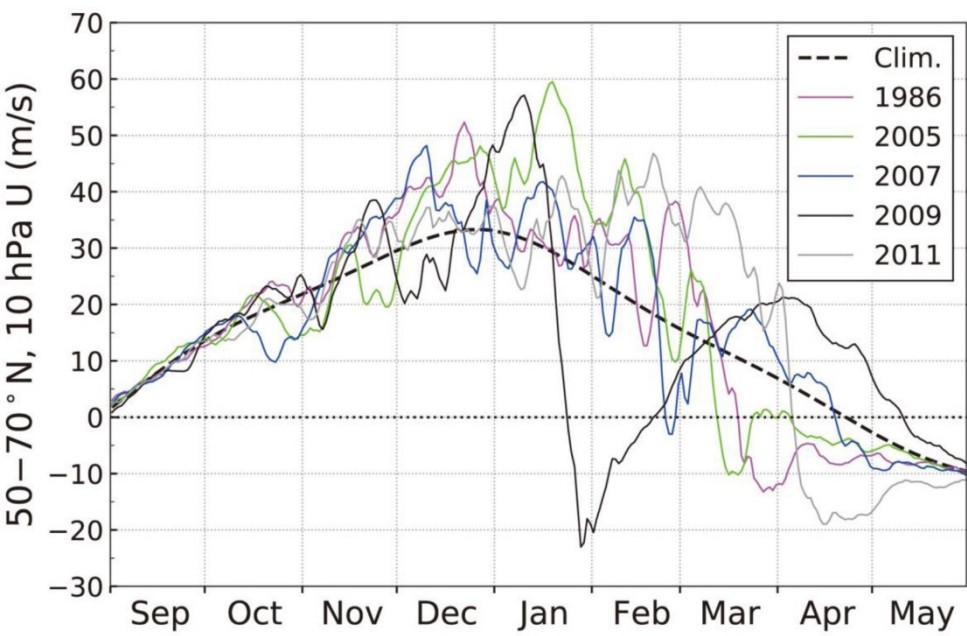

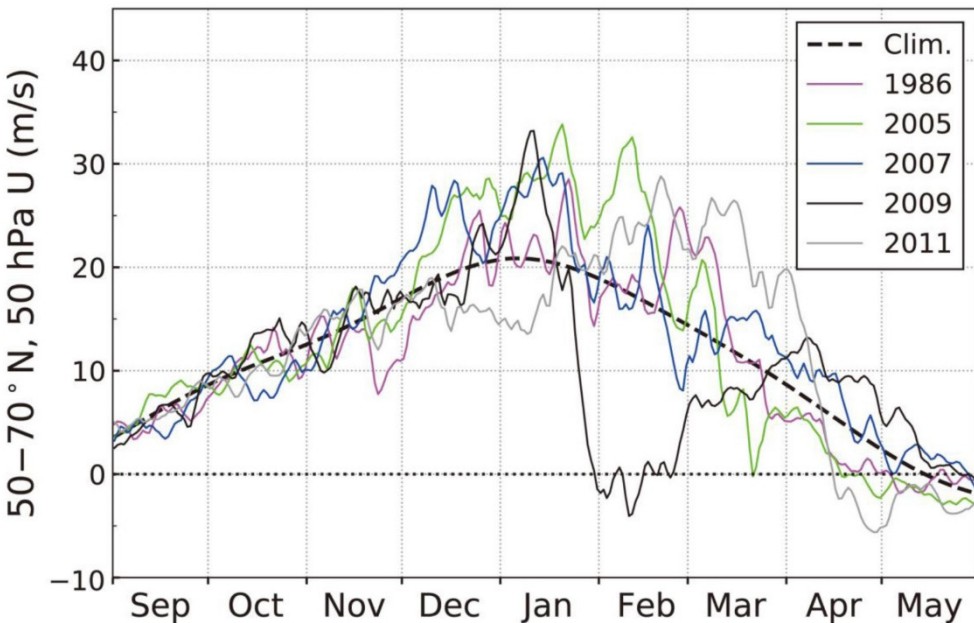

**Figure 5.** Daily time-series of the zonal mean zonal wind over 50−70 N at 10 hPa (upper panel) and 50 hPa (lower panel) from ERA-Interim reanalysis data (solid lines with units of m/s). Purple line: 1986 (September 1985−May 1986), green line: 2005, blue line: 2007, black line: 2009, and gray line: 2011. The broken line denotes the daily climatology of the zonal wind from 1979−2011, smoothed using a Fourier filter with a rectangular window of 3 months.

Regarding the difference in the seasonal evolution of the westerly winds from winter to spring in the QBO and solar phases, Yamashita et al. [10] analyzed the evolution of the polar vortex intensity in the stratosphere in association with the sea level pressure (SLP) and wave activity for December–January. They found that for the QBO-W/$S_{max}$ group, north–south dipole anomalies of SLP were present in the North Atlantic region. Intensification of the positive anomaly was seen from December to January, leading to an enhancement in the planetary wave propagation from the troposphere to the stratosphere,

with a weak polar vortex in late winter. Conversely, for the QBO-W/$S_{min}$ group, north–south dipole anomalies of SLP were not observed, and the planetary wave propagation from the troposphere to the stratosphere was weaker than that for the QBO-W/$S_{max}$ group. This may lead to the persistence of the relatively strong polar vortex in the stratosphere until February–March for the QBO-W/$S_{min}$ group. However, as shown in Figure 5, zonal-mean zonal wind at 10 hPa and 50 hPa in March of the QBO-W/$S_{min}$ group does not show a clear larger magnitude compared to those of other QBO/Solar phases.

Figure 6 indicates anomalies of the monthly mean EP-flux, divergence, residual mean circulation, temperature, and zonal mean zonal wind during the QBO-W/$S_{min}$ years, based on climatology. This figure is a counterpart of Figure 9 for the QBO-W/$S_{max}$ years in Yamashita et al. [10]. Anomalies of the EP-flux, divergence, temperature, and zonal mean zonal wind are similar in December and January between the QBO-W/$S_{max}$ and QBO-W/$S_{min}$ years, whereas they are significantly different in February and March. The easterly anomalies are evident in the Arctic during February and March for the QBO-W/$S_{max}$ years, as indicated by the blue color in Figure 9 of Yamashita et al. [10]. Moreover, the westerly anomalies (red color) in February and March are also evident for the QBO-W/$S_{min}$ years (the panels "i" and "l" in Figure 6). Accordingly, high temperature anomalies (red color) are observed in the Arctic lower stratosphere during February and March for the QBO-W/$S_{max}$ years whereas low temperature anomalies (blue color) are observed during February and March for the QBO-W/$S_{min}$ years (the panels "h" and "k" in Figure 6). The direction of the EP-flux anomalies and the distribution of the EP-flux divergence anomalies in the Arctic stratosphere is similar in December and January; however, it is different in February and March (panels "a", "d", "g", and "j" in Figure 6). These figures indicate that the dynamical fields in February and March are different between the QBO-W/$S_{min}$ years and the QBO-W/$S_{max}$ years, and those in the QBO-W/$S_{min}$ years indicates a stronger polar vortex.

One exception for the QBO-W/$S_{min}$ group is the year 2009. Zonal wind speeds greater than 50 m s$^{-1}$ are apparent in early January 2009, whereas the zonal wind becomes weak and easterly in late January. This indicates a major stratospheric warming (see 2008/2009 in Table 1 and breakdown of the polar vortex. The weak polar vortex continues until February 2009. As the intensity of the polar vortex in 2009 is weak relative to the other four years, the ozone amount in 2009 is exceptionally large for the QBO-W/$S_{min}$ group (Figure 2). As discussed in the next section and Tables 1 and 2, sudden stratospheric warming occurred in 2007 and 2009 of all the QBO-W/$S_{min}$ years. The sudden stratospheric warming in 2007 was categorized as vortex displacement (dominance in wave number one) while that in 2009 was categorized as vortex split (dominance in wave number two) [24]. The effect on the zonal mean zonal wind is smaller in 2007 and much larger in 2009 as shown in Figure 5. Charlton and Polvani [25] showed somewhat larger anomalies of the area-weighted mean 100-hPa polar cap temperature (90°–50° N) and zonal mean zonal wind at 60° N and 10 hPa for the vortex split events than those for the vortex displacement events based on the NCEP-NCAR reanalysis dataset from 1958–2002. They also suggested that vortex displacements and splits should be considered dynamically distinct. However, the reason for such a large influence on the zonal-mean zonal wind in 2009 is not apparent. It is not known if the vortex split was the main cause for the large anomaly in 2009, and whether a relationship exists between the QBO/solar phases and vortex displacement/split.

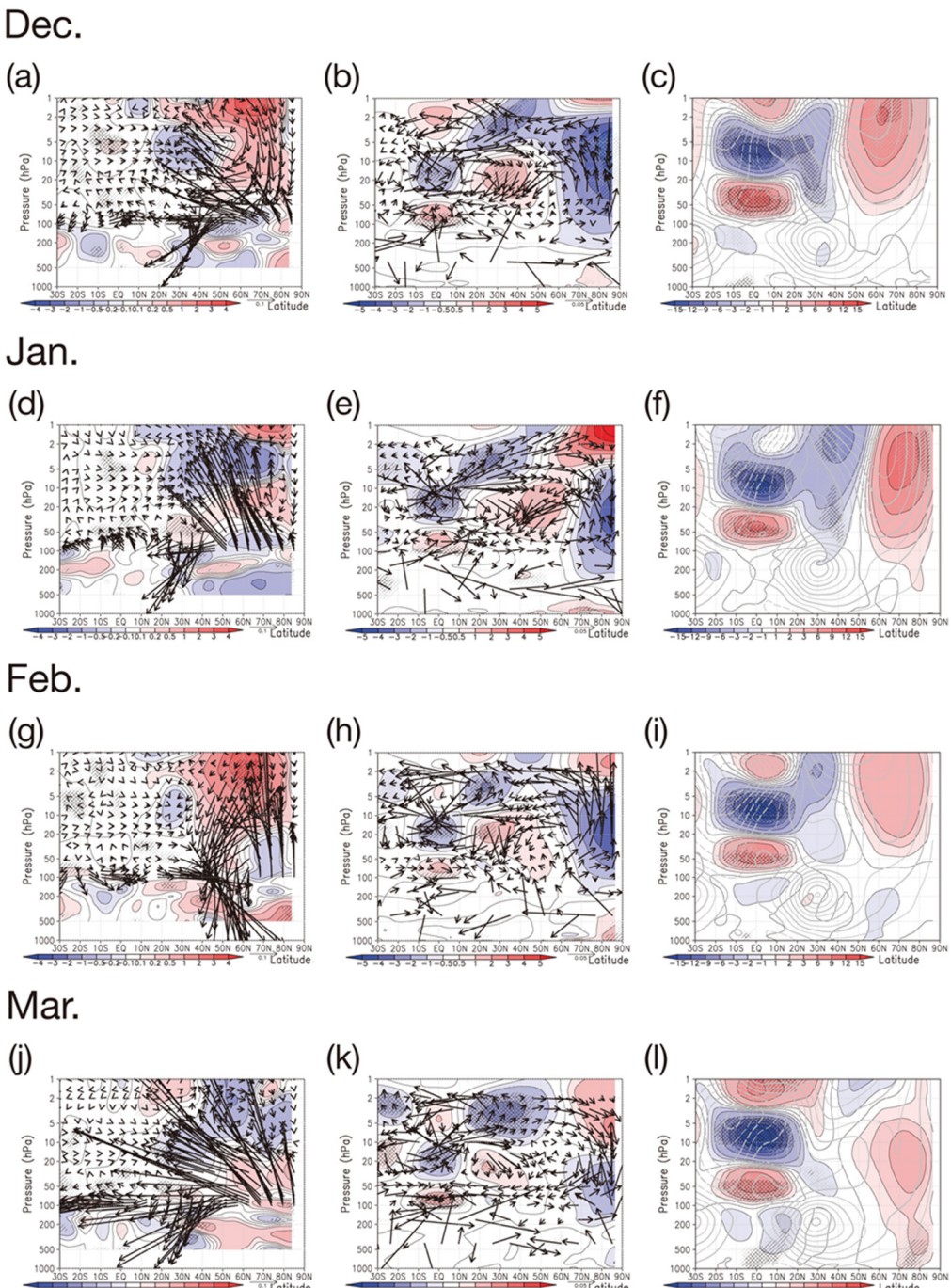

**Figure 6.** (**a**) Longitude–height section of composite anomalies for QBO-W/S$_{min}$ years from the December mean E-P flux (vector) and its divergence (shading, units: m s$^{-1}$ d$^{-1}$) from ERA-Interim. (**b**) Anomalies of the residual mean meridional circulation (vector) and temperature (shading, units: K). The vertical components of the E-P flux and the residual mean meridional circulation are magnified by a factor of 200 relative to the horizontal component, and their scales for the horizontal vector are shown at the bottom right of the panel in units of kg m$^{-1}$ s$^{-2}$ and m s$^{-1}$, respectively. Nine-point smoothing was applied to the gridded data of the E-P flux, its divergence, and the residual mean meridional circulation. (**c**) Anomalies of the zonal-mean zonal wind (shading, units: m/s) with the zonal wind climatology (gray contour, interval is 5 m s$^{-1}$). The hatched areas with gray slash, gray grid, black slash, and black grid denote 80%, 85%, 90%, and 95% statistical significance, respectively, for the anomalies in the E-P flux divergence, temperature, and zonal wind. (**d**–**f**) for January, (**g**–**i**) for February, and (**j**–**l**) for March.

**Table 1.** Central dates of sudden stratospheric warmings (major warmings) in Northern Hemisphere Winters diagnosed from ERA-Interim data.

| NH Winter | Date | QBO/Solar Phase |
|---|---|---|
| 1978/1979 | 22 February 1979 | - |
| 1979/1980 | 29 February 1980 | QBO-E/$S_{max}$ |
| 1980/1981 | 4 March 1981 | QBO-W/$S_{max}$ |
| 1981/1982 | 4 December 1981 | QBO-E/$S_{max}$ |
| 1982/1983 | | |
| 1983/1984 | 24 February 1984 | - |
| 1984/1985 | 1 January 1985 | QBO-E/$S_{min}$ |
| 1985/1986 | | |
| 1986/1987 | 23 January 1987 | - |
| 1987/1988 | 8 December 1987<br>14 March 1988 | - |
| 1988/1989 | 21 February 1989 | QBO-W/$S_{max}$ |
| 1989/1990 | | |
| 1990/1991 | | |
| 1991/1992 | | |
| 1992/1993 | | |
| 1993/1994 | | |
| 1994/1995 | | |
| 1995/1996 | | |
| 1996/1997 | | |
| 1997/1998 | | |
| 1998/1999 | 15 December 1998<br>26 February 1999 | QBO-E/$S_{max}$ |
| 1999/2000 | 20 March 2000 | QBO-W/$S_{max}$ |
| 2000/2001 | 11 February 2001 | - |
| 2001/2002 | 30 December 2001 | QBO-E/$S_{max}$ |
| 2002/2003 | 18 January 2003 | - |
| 2003/2004 | 5 January 2004 | QBO-E/$S_{min}$ |
| 2004/2005 | | |
| 2005/2006 | 21 January 2006 | QBO-E/$S_{min}$ |
| 2006/2007 | 24 February 2007 | QBO-W/$S_{min}$ |
| 2007/2008 | 22 February 2008 | QBO-E/$S_{min}$ |
| 2008/2009 | 24 January 2009 | QBO-W/$S_{min}$ |
| 2009/2010 | 9 February 2010<br>24 March 2010 | - |
| 2010/2011 | | |
| 2011/2012 | | |
| 2012/2013 | 6 January 2013 | - |
| 2013/2014 | | |
| 2014/2015 | 28 March 2015 | QBO-E/$S_{max}$ |
| 2015/2016 | | |
| 2016/2017 | | |

SSWs in the QBO-W/$S_{min}$ are shaded.

**Table 2.** Number of events and winters of sudden stratospheric warmings (major warmings) in the Northern Hemisphere during the four QBO/solar phases.

| | QBO-W/$S_{max}$ | QBO-W/$S_{min}$ | QBO-E/$S_{max}$ | QBO-E/$S_{min}$ |
|---|---|---|---|---|
| 1979–2011 (events) | 3 | 2 | 5 | 4 |
| 1979–2011 (winters) | 3 | 2 | 4 | 4 |
| 1979–2017 (events) | 3 | 2 | 6 | 4 |
| 1979–2017 (winters) | 3 | 2 | 5 | 4 |

### 3.4. Sudden Stratospheric Warming for the Years 1979–2017

During sudden stratospheric warming events in winter and early spring, westerly wind speed of the polar vortex falls, and the temperature rises drastically. Subsequently, chemical ozone loss in the polar vortex would weaken and ozone transport from outside the polar vortex would strengthen, thus increasing the ozone concentration in the polar region. Therefore, sudden stratospheric warming is a factor affecting the ozone amount in the polar region in winter and spring, along with the QBO phase and solar activity, which may be related with each other. We investigate the QBO/solar phase of the years with sudden stratospheric warming. Diagnosis of the occurrence of a sudden stratospheric warming (major warming) event is conducted by the commonly used method using ERA-interim reanalysis data: the central date of the warming is defined as the day when the daily mean zonal mean winds at 10 hPa and 60° N initially change from westerly to easterly between November and March. The winds must return to westerly for 20 consecutive days between the events and the final warmings are excluded [24–26]. This event diagnosis was conducted for the years 1979–2017.

Table 1 presents the central date of stratospheric major warming and the QBO/solar phases from 1979–2017. The central dates diagnosed here are identical with those diagnosed by Butler et al. [26] for the period 1979–2013 and Cohen and Jones [24] for 1979–2010. These studies also used ERA-Interim data for the diagnosis. The period for comparison of the dates with those from previous studies was limited to 2013 by the data availability at that time.

Table 2 presents the number of stratospheric major warming events in the four QBO/solar phases. As the warming occurred twice in the winter of 1998/1999, the number of each warming event and the number of winters with major warmings are shown separately. Additionally, the number is shown for the two different periods; one is for period of the CCM simulation (1979–2011) and the other is that for the total ozone data used in this study (1979–2017). The results indicate a more frequent occurrence of sudden stratospheric warming in the QBO-Easterly years than in the QBO-Westerly years. Although the difference between QBO-W/$S_{max}$ years and QBO-W/$S_{min}$ years is small, sudden stratospheric warming occurred at the least frequency during the QBO-W/$S_{min}$ years. The lesser occurrence of sudden stratospheric warming may be associated with the stable Arctic polar vortex and lesser ozone amount in February and March during the QBO-W/$S_{min}$ years.

### 3.5. Total Ozone Anomaly during the QBO-W/$S_{min}$ Years

Figure 7 presents the equivalent latitude–month section of the total ozone (column ozone) anomaly for the QBO-W/$S_{min}$ group, relative to the average for all years (1979–2011) from December to April. Figure 7a shows a negative ozone anomaly with the magnitude exceeding 20 DU (a negative ozone anomaly of $<-20$ DU) within the equivalent latitudes of 70–90° N during February–March for the TOMS/OMI satellite observations. The total ozone results from the ERA-Interim reanalysis also indicate similar negative ozone anomalies of $<-20$ DU within the equivalent latitudes of 70–80° N in February and 70–90° N in March (Figure 7b). The results of REF-C1SD of the CCM experiment also indicate the occurrence of negative ozone anomalies of $<-20$ DU within the equivalent latitudes of 70–90° N during February–March (Figure 7c), which is in agreement with the TOMS/OMI and ERA-Interim results.

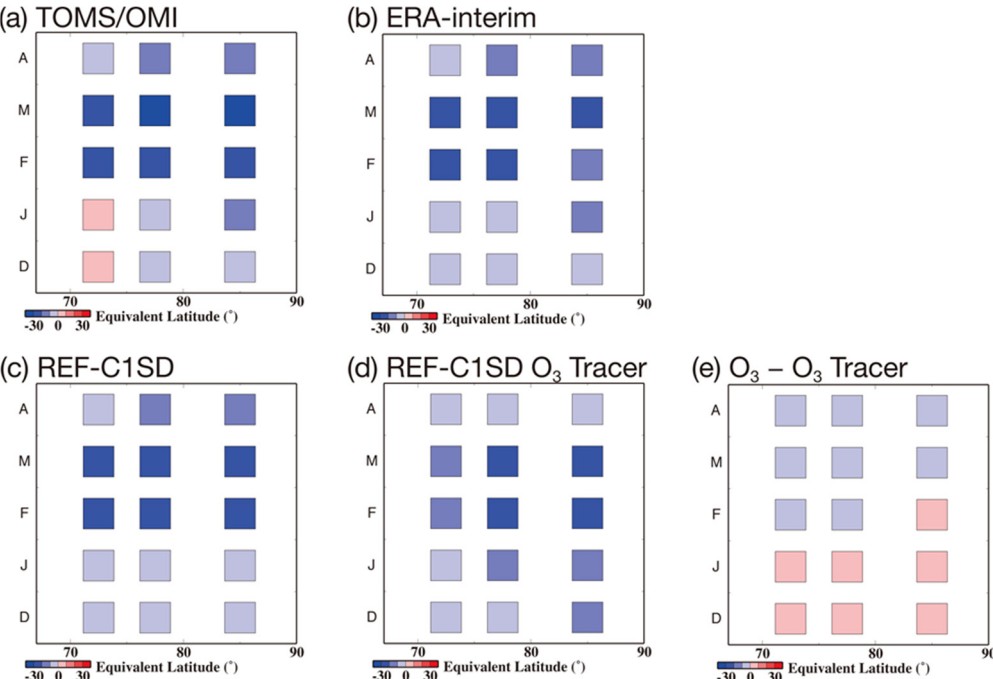

**Figure 7.** (**a**) Equivalent latitude–month section of the total ozone anomaly for QBO-W/$S_{min}$ years from TOMS/OMI data. The horizontal axis indicates equivalent latitude, and the vertical axis denotes the month from December (bottom) to April (top) ("D" denotes December, "J" January, and so on) The interval for the color scale is 10 DU. (**b**) Same as (**a**), but for ERA-Interim data. (**c**) Same as (**a**) but for the REF-C1SD experiment. (**d**) Same as (**c**) but for the total ozone anomaly of the passive ozone tracer. (**e**) Subtraction of the passive ozone tracer anomaly (**d**) from the ozone anomaly (**c**).

To distinguish the effect of the chemical ozone destruction from that of ozone transport for the total ozone anomaly, we analyze the total ozone anomaly of the passive ozone tracer, which is not affected by any chemical change in the CCM (Figure 7d). However, as described in Section 2, it is noteworthy that the ozone tracer is switched off from chemical reactions in the altitude range between 220 hPa and 12 hPa. Thus, the ozone tracer concentration near the boundaries of this altitude range can be affected by chemical reactions at altitudes below 220 hPa and above 12 hPa, where the ozone tracer concentration is identical to the ozone concentration, and is affected by chemistry as well as transport. The total ozone of the passive ozone tracer indicates the presence of negative ozone anomalies, with values of $-10$ to $-20$ DU within the equivalent latitudes of 70–75° N and values of less than $-20$ DU within the equivalent latitudes of 75–90° N in February–March.

Generally, negative anomalies of the passive ozone tracer in February and March are similar in distribution and magnitude to those of chemically active ozone in the REF-C1SD experiment, suggesting that the negative anomalies of the total ozone are mainly caused by ozone transport. This is evident by comparing Figure 7d with Figure 7e, which show the anomaly of the passive ozone tracer and the difference between ozone and the passive ozone tracer (a subtraction of passive ozone tracer concentration from ozone concentration). These figures indicate that the transport effect is dominant compared to the chemical effect in February and March.

Tables 3 and 4 lists values of the total ozone anomaly and partial column ozone anomaly at the three equivalent latitude bands in the polar region and 50–100 hPa for the QBO-W/$S_{min}$ years, respectively. The anomaly of the passive ozone tracer (due to transport) exhibits a larger magnitude with negative values at higher equivalent latitude bands. The chemical anomaly ($O_3$—passive $O_3$ tracer) is largest at the 70–75° N equivalent latitude band in February and at the 75–80° N equivalent latitude band in March, exhibiting negative values at both bands. The change in latitude distribution of the chemical anomaly between February and March may be attributed to the change in sunlight distribution in

the polar region during these months. We observed that the chemical anomaly is less than 6% of the total anomaly in February and 10–20% of the total anomaly in March (Table 3).

**Table 3.** Total ozone anomaly (DU) for the QBO-W/$S_{min}$ years.

| February | | | |
|---|---|---|---|
| Equivalent latitude | 70–75 N | 75–80 N | 80–90 N |
| $O_3$ | −21.2 | −23.5 | −21.2 |
| Passive $O_3$ tracer | −19.9 | −22.7 | −21.9 |
| $O_3$—passive $O_3$ tracer | −1.3 | −0.8 | +0.7 |
| **March** | | | |
| Equivalent latitude | 70–75 N | 75–80 N | 80–90 N |
| $O_3$ | −20.2 | −25.2 | −27.2 |
| Passive $O_3$ tracer | −16.7 | −20.9 | −24.1 |
| $O_3$—passive $O_3$ tracer | −3.5 | −4.3 | −3.1 |

**Table 4.** Partial column ozone anomaly at 50–100 hPa (DU) for the QBO-W/$S_{min}$ years.

| February | | | |
|---|---|---|---|
| Equivalent latitude | 70–75 N | 75–80 N | 80–90 N |
| $O_3$ | −10.3 | −12.1 | −10.9 |
| Passive $O_3$ tracer | −8.1 | −10.4 | −10.6 |
| $O_3$—passive $O_3$ tracer | −2.2 | −1.7 | −0.3 |
| **March** | | | |
| Equivalent latitude | 70–75 N | 75–80 N | 80–90 N |
| $O_3$ | −9.8 | −13.1 | −14.4 |
| Passive $O_3$ tracer | −5.5 | −8.0 | −10.0 |
| $O_3$—passive $O_3$ tracer | −4.3 | −5.1 | −4.4 |

Henceforth, we use the results of the CCM to discuss the vertical profile of ozone. However, it is difficult to validate the vertical profile of the simulated ozone due to the lack of observed ozone vertical profile data, including those over the polar night region in winter. As the ozone amount at 50–100 hPa is greater, the total ozone anomalies during February–March in Figure 7 considerably reflect the ozone anomalies around this altitude range. Thus, we mainly discuss the chemical and dynamical ozone change near 50–100 hPa.

*3.6. Vertical Distribution of the Ozone Anomaly during the QBO-W/$S_{min}$ Years*

The panels in the first and second columns from the left in Figure 8 show the equivalent latitude–height sections of the ozone partial column anomalies at each pressure level of the QBO-W/$S_{min}$ group and that of the passive ozone tracer in January, February, and March. Ozone partial column anomalies is calculated multiplying pressure difference between the top of partial column and the bottom of it by anomaly of ozone mixing ratio of the partial column. There are large negative ozone partial column anomalies of $<-2$ DU within the equivalent latitudes of 70–90° N at 50 and 70 hPa in February. In March, large negative ozone anomalies of $<-2$ DU are observed in the same equivalent latitude ranges at 50 and 70 hPa, and within 80–90° N at 80 and 115 hPa. The partial column of the passive ozone tracer at these pressure levels also shows negative anomalies with slightly smaller magnitudes, especially at 50 hPa in March. Figure 9a–c shows the anomaly of the residual mean circulation for the QBO-W/$S_{min}$ years and the zonal mean ozone mixing ratio averaged for all the years. In the polar lower stratosphere, indicated by the green dotted rectangle, the contours of the ozone mixing ratio are nearly horizontal with the positive vertical gradient. Therefore, a positive anomaly of the vertical component of residual mean circulation may lead to an ozone transport that causes a decrease in the ozone from the average for all years. Moreover, positive anomalies of the vertical wind are evident in the polar lower stratosphere during January–March, which are consistent

with the negative anomalies of the ozone tracer in the polar lower stratosphere during these months in Figure 8. However, a clear difference in the circulation anomaly between February and March is not evident, although a difference in the passive ozone tracer at 50 hPa occurs between February and March. This may partly be the result of depicting the distributions against the geographical latitude, as reflected in Figure 9, rather than against the equivalent latitude, due to the definition of the residual mean circulation on geographical latitude and pressure.

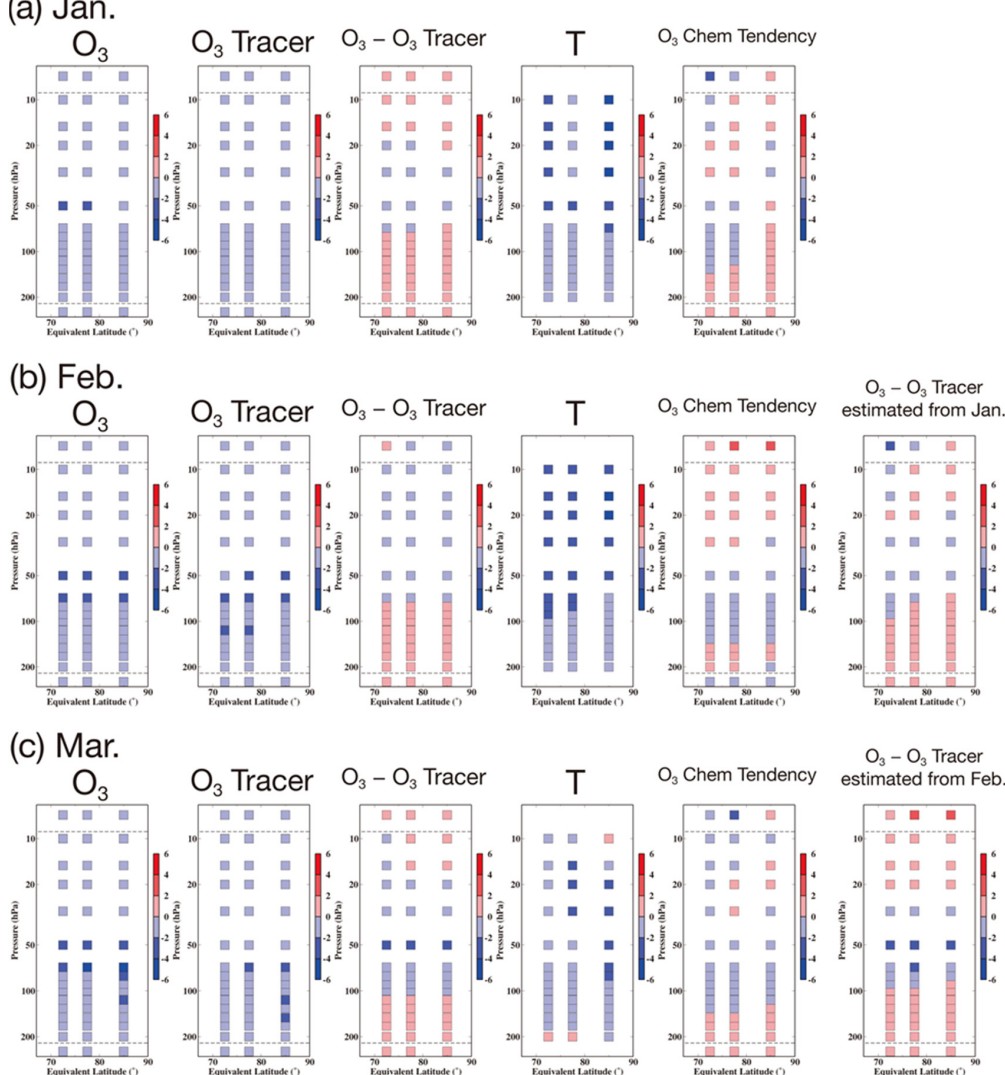

**Figure 8.** (**a**) Equivalent latitude–height section of the partial column ozone anomaly in January (upper left panel) for the QBO-W/$S_{min}$ years from the REF-C1SD experiment. The interval for the color scale is 2 DU. The colored squares above 10 hPa and below 200 hPa indicate the column values above 10 hPa and below 200 hPa, respectively. The 2nd and 3rd panels from the left display the partial column anomaly of the passive ozone tracer and the difference between the ozone anomaly and the anomaly of the passive ozone tracer, respectively. The 4th panel displays the temperature anomaly expressed in the 2 K interval for the color scale, and the 5th panel displays the monthly mean partial column anomaly of the chemical tendency of ozone in units of DU/month with an interval of 2 DU/month for the color scale. (**b**) Same as (**a**) but for February. The 6th panel displays the anomalies estimated by the sum of the monthly mean chemical tendency in January and the difference between the ozone anomaly and the anomaly of the passive ozone tracer in January. (**c**) Same as (**a**) but for March. The 6th panel displays the anomalies estimated by the sum of the monthly mean chemical tendency in February and the difference between the ozone anomaly and the anomaly of the passive ozone tracer in February.

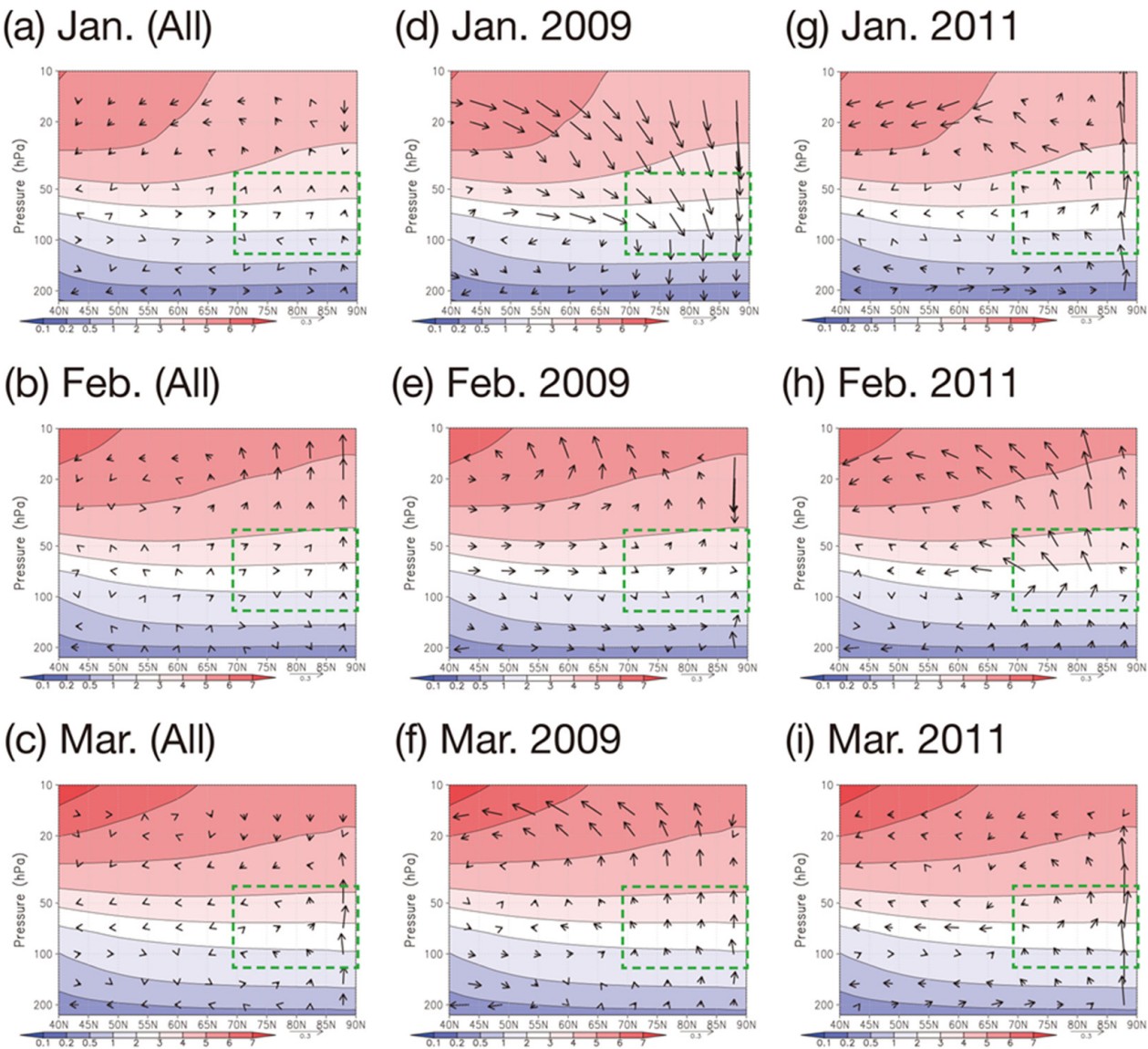

**Figure 9.** Residual mean meridional circulation anomaly for the QBO-W/$S_{min}$ years in (**a**) January, (**b**) February, and (**c**) March. The vertical component of the residual circulation is magnified by a factor of 200 relative to the horizontal component, and the scale for the horizontal vector is shown at the bottom right of the panel in units of m/s. The contours/shadings indicate the ozone mixing ratio averaged for 1979–2011 with units of DU. (**d**–**f**) Same as (**a**–**c**), respectively but for anomalies in 2009. (**g**–**i**) Same as (**a**–**c**) but for anomalies in 2011.

The panels in the third column in Figure 8 indicate the difference in the partial column anomaly between chemical ozone and the passive ozone tracer (i.e., a subtraction of passive ozone tracer concentration from chemically active ozone concentration), which denotes the ozone anomaly created solely by chemistry. The negative anomalies are evident in the lower stratosphere, especially at 50 hPa in March, and the chemical effect is greater than the transport effect at this pressure level. The large negative anomalies in the lower stratosphere at 50–100 hPa in March are associated with the larger negative total ozone anomaly in March than that in February (Tables 2 and 3). However, these chemical effects on the negative ozone anomalies in the lower stratosphere are partly canceled by the positive chemical anomalies below and above the lower stratosphere. This limits the contribution of chemistry to the total ozone anomaly in February and March (Table 3 and Figure 7). Table 4 indicates that the chemical effect in the lower stratosphere (50–100 hPa) is greater in March than in February; however, it is still lower than the transport effect.

The panels in the fourth column in Figure 8 show the temperature anomaly. The negative anomalies between −2 K and −4 K are evident within 70–90° N in the lower and middle stratosphere in February but within a smaller region in March. The panels indicate the negative anomalies in the QBO-W/$S_{min}$ phase in almost all regions at 70–90° N and 10–200 hPa. In regard to chemical ozone change, a negative anomaly in temperature should cause a negative anomaly in ozone in the lower stratosphere where heterogeneous reactions may occur, because additional PSCs are expected to form at lower temperature. Meanwhile, a negative anomaly in temperature should cause a positive anomaly in the ozone at altitudes where only gas-phase chemical reactions occur, because gas-phase chemical reactions produce additional ozone at lower temperatures. The anomalies in the ozone chemical tendency (the fifth column) in February and March are consistent with these anticipated ozone anomalies owing to the gas-phase/heterogeneous reactions through the negative temperature anomaly. In the chemical tendency panels, the anomalies are shown in the units of DU month$^{-1}$. The ozone chemical tendencies in February indicate negative anomalies in the lower stratosphere (50–130 hPa) and positive anomalies below and above these levels. In March, as opposed to February, negative anomalies are evident at 70–75° N above 50 hPa. Anomalies of the surface area of nitric acid trihydrate (NAT), which is the primary component of PSCs in the Arctic, are depicted in Figure 10. Large positive NAT surface area anomalies are evident in the Arctic lower stratosphere. The lager surface area of NAT caused large negative ozone anomalies in the Arctic lower stratosphere during February and March for the QBO-W/$S_{min}$ years, as illustrated in Figure 8.

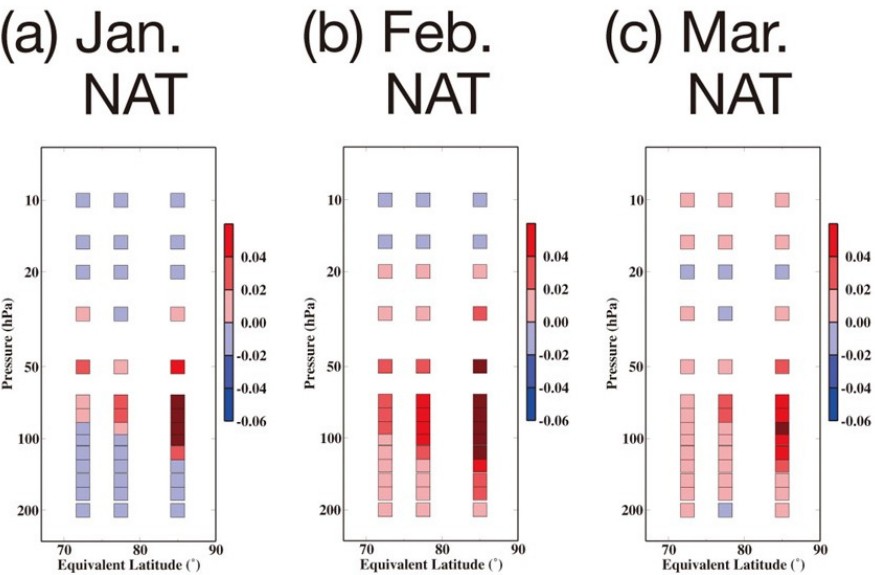

**Figure 10.** Equivalent latitude–height section of the NAT surface area anomaly ($10^{-9}$ cm$^2$ cm$^{-3}$) in (**a**) January, (**b**) February, and (**c**) March for QBO-W/$S_{min}$ from the REF-C1SD experiment.

The panels in the sixth column of Figure 8 highlight the difference in the partial column anomaly between the chemically active ozone and the passive ozone tracer estimated based on the anomalies in the preceding month. The anomalies in February are estimated by adding the monthly mean chemical ozone tendency in January (the top panel in the fifth column in units of DU/month) multiplied by 1.0 month with the difference between the chemically active ozone and the passive ozone tracer in January (top panel in the third column). Similarly, the anomalies in March are estimated based on the middle panels in the fifth and third columns for February. These results indicate that the anomaly distribution of the third column is similar to that of the sixth column, especially at pressure levels of 30–100 hPa. It is confirmed that the differences between the chemically active ozone and the passive ozone tracer in February and March are roughly reproduced by the anomalies and chemical tendency in the preceding month, i.e., January and February, respectively.

### 3.7. Anomalies in the Year 2009 and the Other QBO-W/$S_{min}$ Years

A large zonal mean zonal wind deviation is evident in 2009, compared to the other QBO-W/$S_{min}$ years from mid-January to late February (Figure 5). The positive total ozone anomalies in February and March 2009 are significantly different from the negative anomalies in the other QBO-W/$S_{min}$ years (Figure 2). Therefore, it is necessary to compare the anomalies in 2009 with those in the other QBO-W/$S_{min}$ years.

Figure 11 shows the equivalent latitude–month section of the total ozone anomaly for 2009 from December to April. In contrast to the other QBO-W/$S_{min}$ years (Figure S4), positive anomalies are evident in the polar regions in February and March, as shown in Figure 11a,b. The positive anomaly of the observations (TOMS/OMI and ERA-Interim) is large between 80–90° N. The model reproduced these positive anomalies in 2009, as shown in Figure 11c. The model simulation indicates considerable positive anomalies of total ozone because of the large part of the transport effect in comparison with the chemical effect during this year (Figure 11d,e).

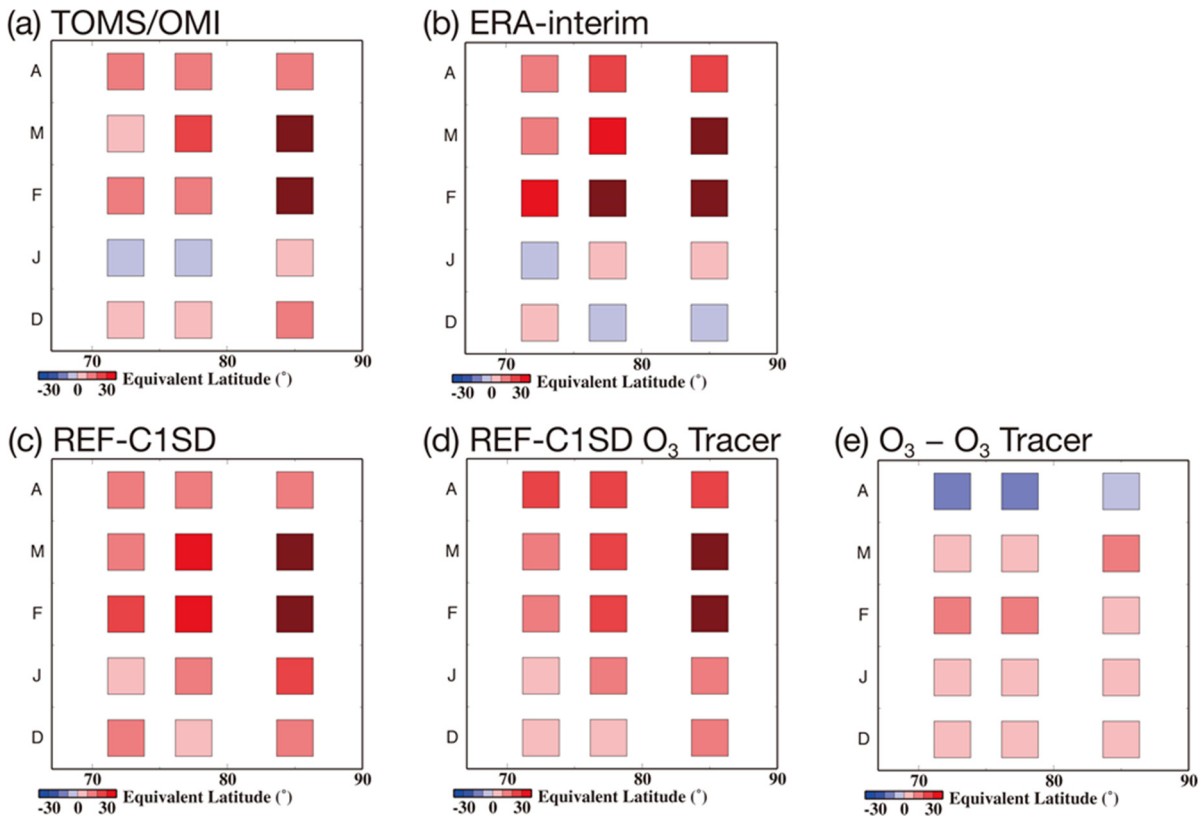

**Figure 11.** Same as Figure 7 but for the anomalies in 2009 (December 2008–April 2009).

The vertical distributions of the ozone partial column anomalies, passive ozone tracer, the difference between ozone and passive ozone tracer, and the temperature from February–March 2009 are shown in Figure 12. The anomalies between Figure 12 and Figure S5 (QBO-W/$S_{min}$ years excluding 2009) are quite different, such as those between Figure 11 and Figure S4. The large zonal mean zonal wind deviation during 2009 compared to the other QBO-W/$S_{min}$ years is evident between mid-January and late February (Figure 5). The westerly wind rapidly decelerated and changed to the easterly wind in late January. Subsequently, the easterly wind decelerated in February and changed back to the westerly wind in late February. This is diagnosed as a major warming (Table 1). The anomaly of the residual mean circulation in January shows a clear poleward and downward direction, and that in February shows a slightly upward direction in the polar lower stratosphere (Figure 9d,e). As shown in Figure 9f, the anomaly in March shows a clear upward and

equatorward direction, which is consistent with the stronger zonal mean zonal wind during this month (i.e., stronger polar vortex than average, see Figure 5). Such differences from the other QBO-W/$S_{min}$ years led to positive anomalies of the passive ozone tracer in the polar lower stratosphere (second column of Figure 12), indicating positive ozone anomalies (first column of Figure 12), which differs appreciably from the other QBO-W/$S_{min}$ years (Figure S6).

The temperature in 2009 exhibits negative and positive anomalies in the polar stratosphere in January and February, respectively. In March, negative and positive temperature anomalies are observed in the polar middle and lower stratosphere, respectively (fourth column of Figure 12). The expected anomalies of the ozone chemical tendency caused by the temperature anomalies are evident in March and in the lower stratosphere in January. In other words, anomalies of the opposite sign between temperature and the ozone chemical tendency in the middle stratosphere and anomalies of the same sign in the lower stratosphere are observed. This relationship is relatively less apparent in February and in the middle stratosphere in January. These weak chemical relationships are consistent with the dominance of the transport effect on the ozone anomaly during this year. Similar to the QBO-W/$S_{min}$ years, the difference between the chemically active ozone and the passive ozone tracer in February and March are reproduced based on the anomalies in the previous month.

The year 2009 shows a considerably different ozone anomaly from the other QBO-W/$S_{min}$ years. The year 2009 shows large positive ozone and temperature anomalies in the lower stratosphere in February and March, which weakens the statistical significance of the negative ozone and temperature anomalies of the QBO-W/$S_{min}$ years.

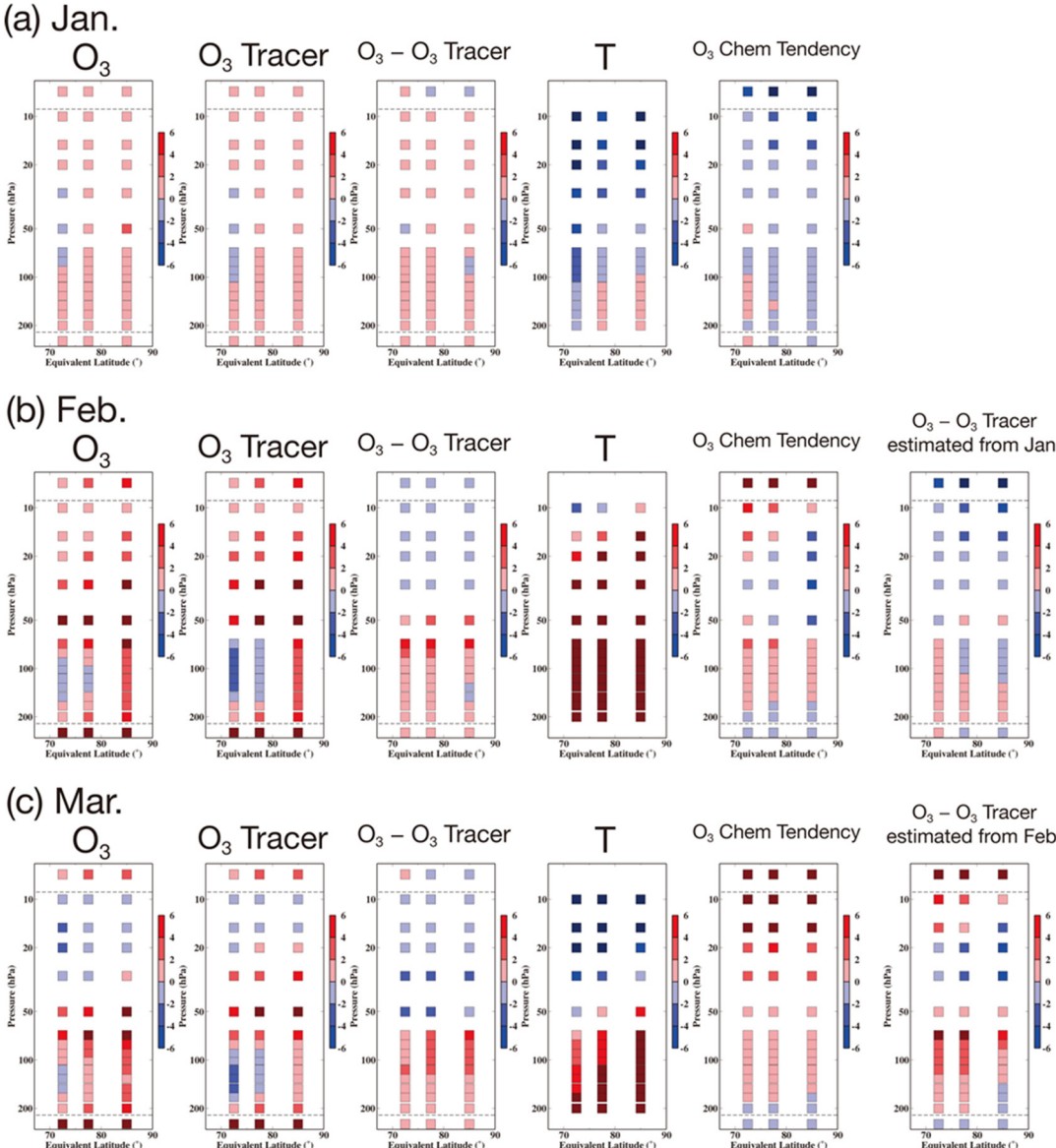

**Figure 12.** Same as Figure 8 but for the partial column anomaly in 2009.

### 3.8. Anomalies in the Year 2011

In March 2011, unusually severe ozone loss occurred in the Arctic region inside the strong polar vortex [27], despite the fact that the concentrations of reactive chlorine ($Cl_y$) and reactive bromine ($Br_y$) in the stratosphere started to decrease around the year 2000. This year is also categorized into the QBO-W/$S_{min}$ group. The large negative total ozone anomalies of approximately $-40$ DU were observed and simulated, particularly in March as well as in February, as shown in Figure 13a–c. Figure 13d,e indicate a large negative transport anomaly and larger negative chemical anomaly in March 2011 than in February 2011. The chemical effect on the total ozone anomalies for 2011 is larger than that for the other QBO-W/$S_{min}$ years, with values between 6% and 20% of the total effect in February and between 20% and 34% in March. The values for March 2011 are consistent with the results by Isaksen et al. [28], which reports 23% of the chemical anomaly to the total ozone anomaly during this month.

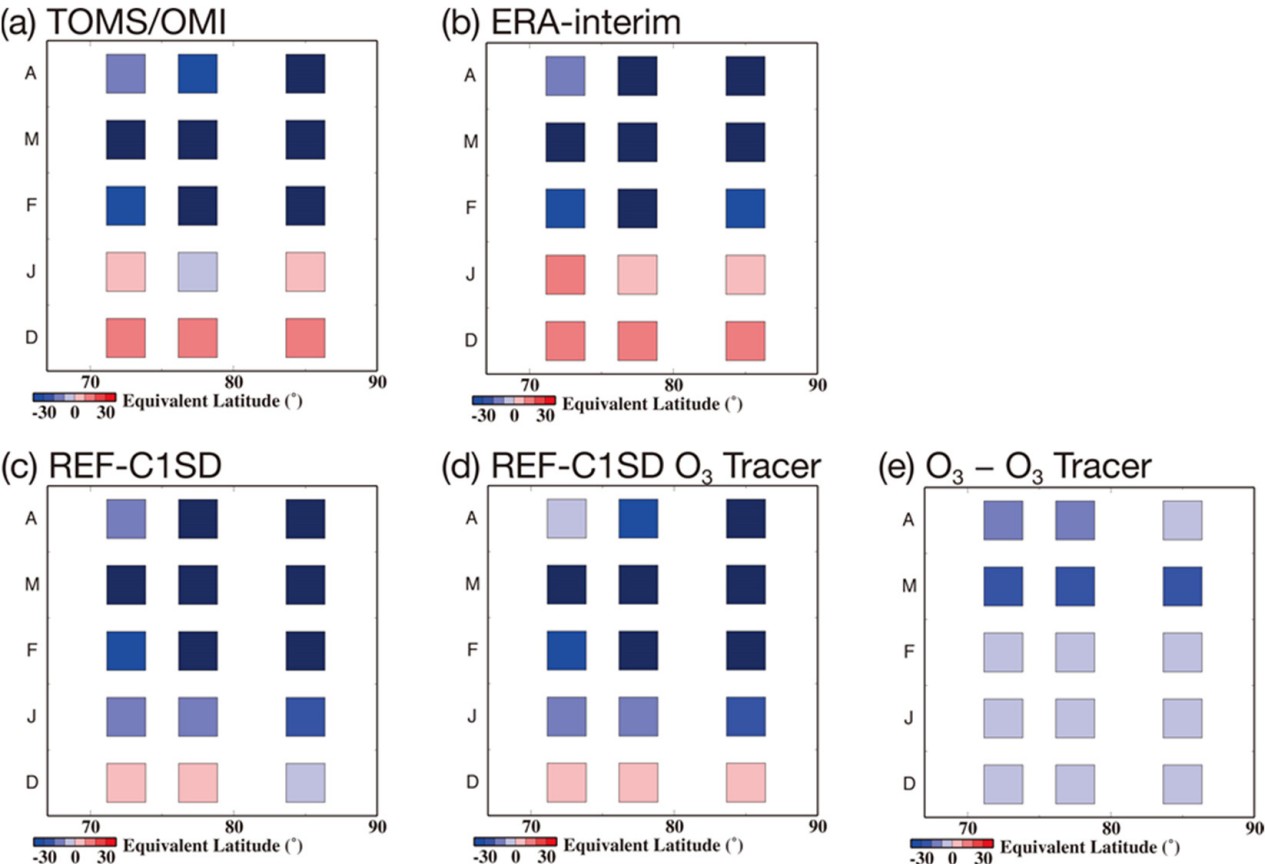

**Figure 13.** Same as Figure 7 but for the anomalies in 2011 (December 2010–April 2011).

Figure 14 shows considerably large negative transport anomalies in the polar stratosphere in February and March (the middle and bottom panels of the second column). The negative transport anomalies are consistent with the residual mean circulation anomalies in the polar stratosphere, which show positive anomalies of the vertical wind (Figure 9h,i). The bottom panel of the third column indicates large negative chemical anomalies within the equivalent latitudes of 70–90° N between 50 and 80 hPa, which resulted in the large negative total ozone anomalies in March, as shown in Figure 13. The temperature shows significantly large negative anomalies in the polar stratosphere in February and March. The chemical ozone tendency anomalies in February and March (the middle and bottom panels of the fifth column) show the same sign as that of the temperature anomaly in the lower stratosphere and the opposite sign in the middle stratosphere, as expected from the temperature dependence of the chemical ozone production/destruction in these regions. Finally, as shown in the middle and bottom panels of the sixth column, the difference between the chemically active ozone and the passive ozone tracer in February and March are reproduced by the anomalies in the previous months, i.e., January and February, respectively, especially at the levels of 30–100 hPa.

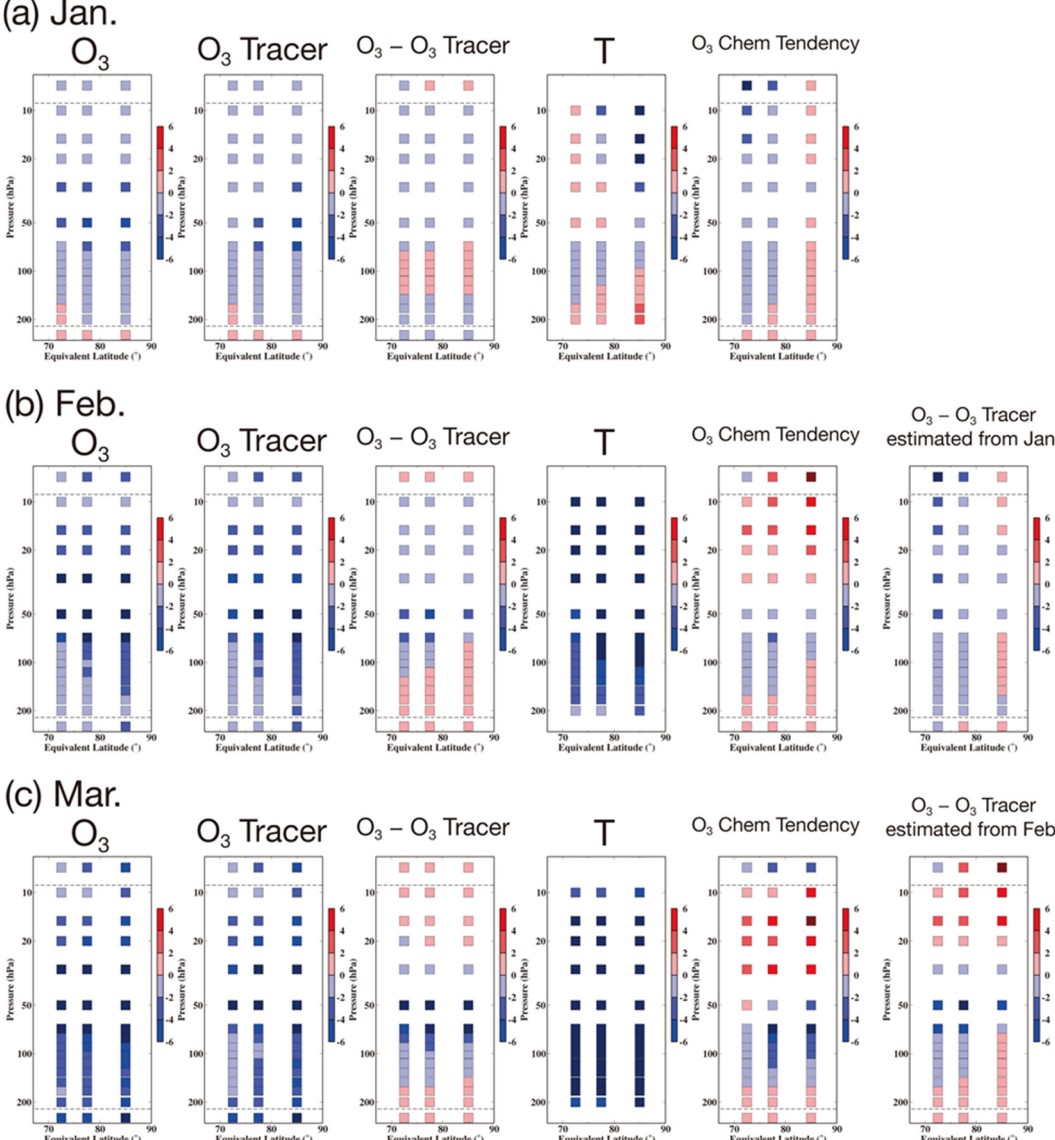

**Figure 14.** Same as Figure 8, but for the partial column anomaly in 2011.

## 4. Summary

In this study, we investigate the ozone and associated dynamical quantities mainly for the years categorized into the phase of QBO-westerly (QBO-W)/solar minimum ($S_{min}$) for 1979–2011. The objective of this study is examining the relationship between the phases of the QBO/11-year solar cycle and total ozone anomaly in late winter and early spring associated with the strength of NH polar vortex, along with the large interannual variation in the Arctic region.

Our analyses suggest that the total ozone amount in March averaged for the QBO-W/$S_{min}$ years (1986, 2005, 2007, 2009, and 2011) is slightly smaller than those averaged for the QBO-W/$S_{max}$ and QBO-E/$S_{max}$ years, whilst it is difficult to say that the zonal-mean zonal wind in March averaged for the QBO-W/$S_{min}$ years is stronger than those of the other phases in consideration of the standard deviations. Because the total ozone and zonal-mean zonal wind in March averaged for the QBO-E/$S_{min}$ years have very large standard deviations, comparisons of this phase with the other phases are difficult.

The QBO-W/$S_{min}$ years excluding 2009 exhibited a lower total ozone at 70–90° N in March than the average for 1979–2011. More recently, in 2017, which is also categorized

into the QBO-W/$S_{min}$ group, a low total ozone amount during March was exhibited in the OMI data but not in the ERA-Interim data.

The dynamical fields, such as the EP-flux, its divergence, temperature, and zonal mean zonal winds, show a stronger NH polar vortex and colder temperature in the Arctic from winter to early spring during the QBO-W/$S_{min}$ years compared to the climatology.

Our analysis of the ERA-Interim data indicates that a sudden stratospheric warming occurred at the lowest frequency in the QBO-W/$S_{min}$ years. The minimal occurrence of this sudden stratospheric warming during the QBO-W/$S_{min}$ years may be associated with the stable Arctic polar vortex and lower ozone amount in February and March.

The total ozone from TOMS/OMI and ERA-Interim in the QBO-W/$S_{min}$ group showed lower than average values in February and March at the equivalent latitude range of 70–90° N. The CCM satisfactorily simulates the negative anomalies in the total ozone during spring in the polar regions. We also analyzed the total ozone anomaly for the passive ozone tracer. The results indicate that the negative anomaly in total ozone is mostly caused by transport. The chemical anomaly (estimated by subtracting the passive ozone tracer concentration from the ozone concentration, i.e., $O_3$—$O_3$ tracer) is less than 6% of the total anomaly in February and between 10% and 20% in March.

In addition to the total ozone, we also examined the vertical distribution of the partial ozone column anomaly using the CCM output. During the QBO-W/$S_{min}$ years, large negative partial ozone column anomalies of $<-2$ DU were observed within the equivalent latitudes of 70–90° N at 50 and 70 hPa in February. In March, the region of negative ozone anomalies of $<-2$ DU extended slightly into lower altitudes.

In February, the anomalies in the passive ozone tracer and residual mean circulation suggest that the negative anomalies in the polar lower stratosphere are largely caused by the transport effect. The positive anomaly in the vertical component of the residual mean circulation in the polar lower stratosphere is associated with a stronger and more stable polar vortex. The transport anomaly is also large in March; however, it is smaller than that in February.

Among the QBO-W/$S_{min}$ years, in 2011, when an unusually severe ozone loss occurred in the Arctic, very large negative anomalies occurred in February and March, and evidence of the dominance of the transport effect was seen. The chemical effect on the total ozone anomalies was between 6% and 20% of the total effect in February and between 20% and 34% in March.

These results suggest that the ozone in the Arctic spring tends to be lower during the QBO-W/$S_{min}$ years than during the other years owing to the dominance of the transport effect. It is plausible, though not statistically significant, that the reduced occurrence of SSW in QBO-W/$S_{min}$ condition led to the stronger polar vortex and the weaker downward residual motion in the polar lower stratosphere in late winter to early spring, which results in the lower ozone in Arctic spring. However, the statistical significance of the anomaly is decreased by one of the QBO-W/$S_{min}$ years (i.e., 2009), which exhibits a different seasonal evolution than other years in the QBO-W/$S_{min}$ group. This is attributed to the sudden stratospheric warming, dominated by wave number two, indicating large positive anomalies in the ozone and temperature in the Arctic region. Moreover, the small sample size of the QBO-W/$S_{min}$ years for 1979–2011 was also a factor that lowered the confidence of our results, although the negative anomaly in the OMI total ozone in March 2017 during the QBO-W/$S_{min}$ phase may improve the confidence of our results. Furthermore, Kren et al. [11] suggested that the apparent relationship among the QBO, solar cycle, and state of the NH polar vortex can arise from sampling over relatively short periods as considered in this study. Therefore, additional future studies based on a larger sample size of the QBO-W/$S_{min}$ years, including extended past and future years, are needed. Also, it is important to observe whether or not the occurrence of SSW is reduced in the QBO-W/$S_{min}$ condition in association with Arctic ozone variation.

**Supplementary Materials:** The following are available online at https://www.mdpi.com/article/10.3390/atmos12050582/s1, Figures S1–S6: Supporting Information for "Analysis of Arctic spring ozone anomaly in the phases of QBO and 11-year solar cycle for 1979–2017" by Yousuke Yamashita, Hideharu Akiyoshi, and Masaaki Takahashi.

**Author Contributions:** Conceptualization, H.A.; Data curation, Y.Y.; Formal analysis, Y.Y.; Funding acquisition, Y.Y. and H.A.; Methodology, Y.Y. and H.A.; Supervision, H.A. and M.T.; Validation, Y.Y., H.A. and M.T.; Visualization, Y.Y. and H.A.; Writing—original draft, Y.Y.; Writing—review and editing, H.A. and M.T. All authors have read and agreed to the published version of the manuscript.

**Funding:** This research was funded by the Environment Research and Technology Development Fund (2-1303 and JPMEERF20172009) of the Ministry of the Environment, Japan, the GRENE Arctic Climate Change Research Project, Ministry of Education, Culture, Sports, Science, and Technology (MEXT) of Japan, and Japan Society for the Promotion of Science (JSPS) KAKENHI Grant Number JP16K16186, JP16H01183, JP19K03961, JP20K12155, and JP20H01977.

**Institutional Review Board Statement:** Not applicable.

**Informed Consent Statement:** Not applicable.

**Data Availability Statement:** The ERA-Interim dataset used in this study can be accessed from the ECMWF website (http://www.ecmwf.int/, accessed on 28 April 2021). The Generic Mapping Tools (GMT), Matplotlib, and the Grid Analysis and Display System (GrADS) were used to draw the figures. ERA-Interim reanalysis data is available at http://apps.ecmwf.int/datasets/ (accessed on 28 April 2021). Total ozone data of TOMS and OMI are available at https://disc.gsfc.nasa.gov/datasets/TOMSN7L3dtoz_V008/summary?keywords=ozone%20 (accessed on 28 April 2021) and https://disc.gsfc.nasa.gov/datasets/TOMSEPL3dtoz_V008/summary?keywords=ozone%20 (accessed on 28 April 2021). The REF-C1SD simulation data are stored at the CCMI site of CEDA archive at http://data.ceda.ac.uk/badc/wcrp-ccmi/data/CCMI-1/output/NIES (accessed on 28 April 2021).

**Acknowledgments:** The authors thank Nagio Hirota in NIES, Toshihiko Hirooka in Kyushu University, Fumio Hasebe in Hokkaido University, and Makoto Inoue in Akita Prefectural University for their helpful discussions and comments. The CCM calculations were performed using a supercomputer system (NEC-SX9/A(ECO)) at the Center for Global Environmental Research (CGER), National Institute for Environmental Studies (NIES).

**Conflicts of Interest:** The authors declare no conflict of interest. The funders had no role in the design of the study; in the collection, analyses, or interpretation of data; in the writing of the manuscript, or in the decision to publish the results.

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
