# Peer review of "Analysis of Arctic Spring Ozone Anomaly in the Phases of QBO and 11-Year Solar Cycle for 1979–2017"

_atmosphere, doi:10.3390/atmos12050582_

Round 1
Reviewer 1 Report
Comments on the manuscript entitled “Analysis of Arctic spring ozone anomaly in the phases of QBO and 11-year solar cycle for 1979–2011” by Yousuke Yamashita et al.
This paper investigates the influence of the quasi-biennial oscillation (QBO) of the equatorial stratosphere and the 11-year solar cycle on the Arctic spring ozone with the aid of a chemistry climate model as well as satellite data and reanalysis field. It is found that Arctic total ozone in March was slightly smaller in the QBO-westerly (QBO-W)/solar minimum (Smin) condition than in the average of QBO-W/Smax and QBO-E/Smax conditions. This negative anomaly is consistent with the weakening of downward residual motion in the polar lower stratosphere, while chemical processes contribute about 10-20 % of the anomaly. The lower ozone in Arctic spring during QBO-W/Smin years is associated with stronger Arctic polar vortex from late winter to early spring, which is linked to the reduced occurrence of stratospheric sudden warming (SSW). This paper carefully examines the relationship between QBO/solar activity and the spring-time Arctic ozone anomalies and quantifies the contribution of dynamical and chemical processes leading to such anomalies. Although the results are not statistically significant due mainly to the small number of samples, I would like to recommend that the paper be published after some comments given below are adequately considered.
Major comments
1. Some quantitative description is required rather than using the term “slightly smaller” to express the findings that Arctic total ozone in March was smaller in QBO-W/Smin condition than in the average of QBO-W/Smax and QBO-E/Smax conditions. It should be expressed something like, “Arctic total ozone in March is ~ xx DU (~ yy %) smaller in QBO-W/Smin condition than in the average of QBO-W/Smax and QBO-E/Smax conditions, although it is not statistically significant at the 68 % confidence level (1 sigma).”
2. I understand the authors carefully avoid explicit mentioning of the cause-and-effect relationship. But the arguments could to be clearer if written along the sequence of time. Can we understand “it is plausible, though not statistically significant, that the reduced occurrence of SSW in QBO-W/Smin condition leads to the stronger polar vortex and the weaker downward residual motion in the polar lower stratosphere in late winter to early spring, which results in the lower ozone in Arctic spring most pronounced in March”? If not, which link is missing? If so, we may well say that, as the amplification of the planetary waves in the stratosphere is not totally controlled by QBO and F10.7, the exceptional occurrence of the vortex-split type SSW in 2009 disrupted the statistical relationship described above. Then the question is, why the occurrence of SSW is reduced in QBO-W/Smin condition, which is left for future studies.
3. There found some confusion in the citations of section, references, and figures as is detailed below. It makes me hard to complete a close review of the manuscript.
Specific comments
L20: Which is better, “stronger” or “more stable”?
L46, L625: Karen et al. [11] will be Kren et al. [11].
L145: considerably -> largely
L173: Equivalent latitude appears earlier in L90. Reference of Nash et al. (1996) should be made here in L90, which will be accompanied by renumbering of [23].
L203-204: “comparisons of the total ozone amount among the phases may be difficult”
Consider a possible rewording, “it is hard to find significant differences in the total ozone amount among the categories”
Figure 3: The vertical integration to derive total ozone may have obscured the differences among categories. In view of a large negative partial ozone column anomalies at 50 and 70 hPa in February and March (Fig. 8), more significant results could be obtained if similar comparisons are made on the ozone concentrations averaged poleward of 70N equivalent latitude on single isentropic surfaces such as 430 K, 450 K, and 475 K.
L239: Section 4.1: Is it Section 3.1?
Figure 4: It is also interesting to make similar comparisons in the residual mean vertical velocities averaged over 70-90N.
L252, L268, L272: Yamashita et al. [15]: Is it Yamashita et al. [10] or what else?
L311: yeas ?
L322: Butler et al. (2017) -> Butler et al. [26]; Cohen and Jones (2012) -> Cohen and Jones [24]
L428: Figure 9(a-c): There is a confusion; Figure 9 is not the anomaly of the residual mean circulation but NAT. Do you mean Figure 12?
L514-516: A proper explanation of Figure 12 is necessary. If the citation of Figure 9(a-c) in L428 is a confusion of Figure 12, the numbering of figures needs to be corrected.
L520: Figure 12: Is it Figure 11?
L551: Isaksen et al. (2012) is missing in the references.
Reviewer 2 Report
General comments.
- The title does not accurately reflect the content of the article. In fact, the article analyzes the period from 1979 to 2017, although the results of numerical simulations are really only available until 2011. It is recommended to reflect the period up to 2017 in the title.
- Figure 1. It looks like 2016 year can also be attributed to QBO-W/Smin. Both the zonal wind and solar activity indicators for 2016 almost coincide with the indicators for 2011.
- To analyze the stability of the polar vortex, it would be useful, along with the average characteristics for March, to consider the characteristics averaged over February (Fig. 2-4).
- Figure 5. The bottom panel is announced in the caption, but is not shown in the figure. Meanwhile, a 50 mb zonal wind variability might be even more useful than its variability at 10 mb. The same is true for Fig. 4.
Partial comments.
- Lines 94-94. It is desirable to indicate the upper boundary of the model.
- Lines 576-580. Comments on QBO-E/Smin is missed.
- The pictures in the figure 9 are too small, although there is space to enlarge them.
Reviewer 3 Report
Review for Yamashita et al.,
"Analysis of Arctic spring ozone anomaly in the phases of QBO and 11-year solar cycle for 1979–2011"
submitted to the Atmosphere
Comments:
Here authors revisit widely accepted QBO dependent relationship between 11-year solar cycle variability and Artic ozone.
Authors use combination of satellite, reanalysis data sets as well as outputs from a CCMs
for 1979–2011 time period. Authors find that the composite mean
of the NH high-latitude total ozone in the QBO-westerly phase during solar minima are slightly smaller
than those averaged for the QBO-W/Smax and QBO-E/Smax years.
They also analyse CCM simulated passive ozone to shoat that that negative ozone
anomaly during QBOW/Smin is primarily caused by transport. The negative anomaly is consistent with a weakening of
the residual mean downward motion in the polar lower stratosphere. Their analysis suggests that the chemical
processes estimated using the column amount difference between ozone and the passive ozone
tracer is between 10–20% of the total anomaly in March.
Overall this is well written clear and concise manuscript and I would like to recommend it for the publication in a present form.
